# Rethinking CNN's Generalization to Backdoor Attack from Frequency Domain

**Quanrui Rao[1,2], Lin Wang[3] & Wuying Liu[1]** [*]

[1] Shandong Key Laboratory of Language Resources Development and Application, Ludong University, China
[2] School of Information and Electrical Engineering, Ludong University, China
[3] Xianda College of Economics and Humanities, Shanghai International Studies University, China
`quanruirao@m.ldu.edu.cn, lwang@xdsisu.edu.cn, wyliu@ldu.edu.cn`

## Abstract

Convolutional neural network (CNN) is easily affected by backdoor injections, whose models perform normally on clean samples but produce specific outputs on poisoned ones. Most of the existing studies have focused on the effect of trigger feature changes of poisoned samples on model generalization in spatial domain. We focus on the mechanism of CNN memorize poisoned samples in frequency domain, and find that CNN generate generalization to poisoned samples by memorizing the frequency domain distribution of trigger changes. We also investigated the assistance provided by perturbations generated at different frequencies to the generalization, and explore the influence of trigger perturbations in different frequency domain components on the generalization of poisoned models from visible and invisible backdoor attacks. We prove that high-frequency components are more susceptible to perturbations than low-frequency components. Based on the above fundings, we propose a universal invisible strategy for visible triggers, which can achieve trigger invisibility while maintaining raw attack performance. We also design a novel frequency domain backdoor attack method based on low-frequency semantic information, which can achieve 100% attack accuracy on multiple models and multiple datasets, and can bypass multiple defenses.

## 1 Introduction

Convolutional Neural Network (CNN) excel at multi-domain tasks, but their instability and susceptibility make them highly sensitive to small perturbations. As a result, the security of CNN has been widely concerned, among which the backdoor attack is an attack method that utilizes the instability of CNN (Tang et al. (2021)). A backdoor attack happens when an attacker control data or model parameters during the training phase to inject a backdoor into the model. After that, the model behaves normally on the original samples but can be made to generate specific outputs on the poisoned samples during the inference phase (Saha et al. (2020); Yao et al. (2019); Costales et al. (2020)). Some scholars have investigated the mechanism of CNN memorize poisoned images (Li et al. (2020); Datta & Shadbolt (2022)), but most studies only examined the effect of changing a specific trigger feature in the spatial domain on the model's generalization. The limitations of the spatial domain make it difficult to understand how CNNs generate generalization on poisoned samples.

Frequency domain analysis provides a new perspective on better understanding the mechanism of backdoor attacks. Several scholars have explored the frequency domain generalization performance of CNN. Among them, the work of Wang et al. (2020) has inspired us. They believe that humans associate each image with a corresponding label based on low-frequency information, so the low-frequency component of the image is strongly correlated with the label. In this scenario, when low-frequency information is artificially correlated with labels, the CNN will preferentially memorize the low-frequency components to minimize training loss, and then gradually memorize the high-frequency components in order to achieve higher training accuracy. For this idea, Luo et al. (2019) rigorously derived the same conclusion through mathematical formulas.

---

[*]Corresponding Author

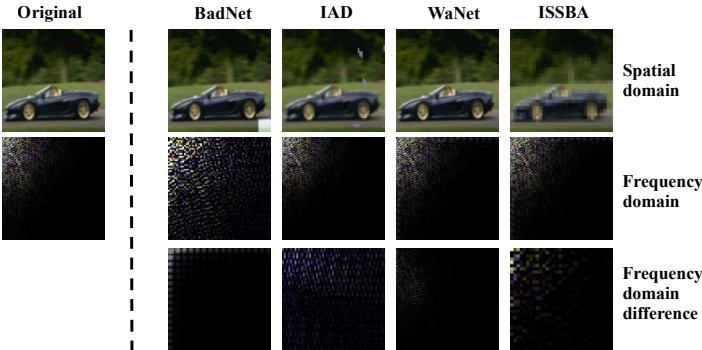

Figure 1: The frequency domain disturbance generated by visible and invisible triggers. The first row shows images in the spatial domain, the second row shows images transformed into the frequency domain using DCT, and the third row shows frequency domain disturbances generated by triggers.

In this paper, we explore the mechanism of memorization in CNN for poisoned images from the perspective of frequency domain, using visible and invisible trigger induced perturbations as examples. Due to the labels on poisoned images are specific, their low-frequency and high-frequency components are treated equally during CNN training. To achieve generalization, the model needs to memorize specific frequency-domain distributions from poisoned images, but its small number and complex semantic information make this process difficult. The injection of triggers unifies the frequency-domain distribution of images from distinct categories, prompting CNN to memorize the distribution so as to generate generalization. CNN can study a specific frequency domain feature distribution and establish the association between feature distribution and labels to achieve backdoor attacks even if the triggers are diverse or not fixed. We went even further to investigate the assistance provided by perturbations generated by visible and invisible triggers at different frequency components to the generalization capability of CNNs. We found that for some triggers, only a subset of perturbations tends to be effective, while for others, efficient attacks can be achieved with only a fraction of the perturbation. Additionally, We examined the sensitivity of different frequency components to trigger perturbations. It is proven that the low-frequency components are more difficult to be affected by perturbations due to their strong correlation with the original tags, whereas high-frequency components are more susceptible to perturbations and have higher attack efficiency due to their weak correlation with the original tags.

Visible backdoor attacks have a higher success rate, but their usability is limited in scenarios with high requirements for stealth due to the visibility of their triggers. To expand the applicability of visible triggers, we propose a universal strategy to conceal visible triggers. This strategy extracts perturbations induced by triggers in middle and high-frequency components, effectively reducing their visibility while maintaining attack effectiveness. In order to efficiently implement backdoor attacks on target categories while minimizing damage to the original image energy, we also propose a backdoor attack algorithm based on low-frequency semantic information. It generates triggers using partial low-frequency components of the target class samples and injects them into the middle and high-frequency components to create poisoned samples. This approach enables 100% attack accuracy while maximizing the visibility of the triggers.

**Our contributions.** Our key contributions in this paper are as follows: **(1)** We explore the mechanism of CNN memorization for poisoned images from a frequency domain perspective, investigating the generalization of CNN with respect to perturbations in different frequency domains. **(2)** We explore the generalization of CNN for visible and invisible triggers, demonstrating that high-frequency features are more susceptible to perturbations than low-frequency features. **(3)** We propose a generalized strategy for rendering visible backdoor attacks invisible while maintaining algorithmic performance. **(4)** We propose a backdoor attack algorithm based on low-frequency semantic information for target classes, achieving high success rates across diverse datasets and models.

## 2    RELATED WORK

### 2.1    BACKDOOR ATTACKS

The concept of backdoor attack was firstly proposed by Gu et al. (2017), who firstly set up the scenario and described the fundamental processes. Specifically, attackers use triggers to inject back-

doors during the model training phase. The poisoned model behaves similarly to the clean model for inputs without triggers. Once a trigger is added to the input, the poisoned model is incorrectly directed to perform a subtask specified by the attacker (Quiring & Rieck (2020)).

Backdoor attacks can be categorized as visible backdoor attacks or invisible backdoor attacks based on the visibility of the trigger. Visible backdoor attacks usually use obvious trigger patterns. For example, Nguyen & Tran (2020a) use a conditional trigger generator to generate unique visible triggers based on input images. This mode can achieve high attack accuracy with fewer poisoned samples, but it is easy to be detected in the process of manually examining training samples. As a result, the concept of invisible backdoor attacks is proposed. Invisible backdoor attacks pay more attention to the process of embedding triggers. For instance, Nguyen & Tran (2020b) achieve trigger unseen by distorting images, and Li et al. (2021b) embed string in images using pre-trained encoder-decoder networks. Additionally, there exists another type of attacks that make it difficult for observers to notice the presence of triggers by combining them with the real scene. This type of backdoor attack focuses more on physical practicality, combining triggers with natural behavior of target systems or real scenes, rendering the presence of triggers inconspicuous. This strategy of attack is also known as a physical backdoor attack (Sarkar et al. (2022)). In this paper, the visible and invisible backdoor attack we study mainly refers to whether the trigger can be seen. We analyze the mechanisms of backdoor attacks according to the distinct impact of frequency-domain perturbations in visible and invisible backdoor attacks on the model.

## 2.2 FREQUENCY DOMAIN RESEARCH ON DNN

The uninterpretable nature of deep neural network (DNN) makes it difficult for people to understand how DNN generate generalization. The F-Principle proposed by Luo et al. (2019) proves that neural network generalization comes from paying more attention to low-frequency components during training. There is also a sequence from low frequency to high frequency in the process of fitting the objective function. Inspired by the F-Principle, Zhang et al. (2019) proposed an efficient Linear F-Principle (LFP) dynamics model. The LFP model establishes the relationship between differential equations and neural network by using differential equations to explain and approximate the training behavior of complex neural network, thus providing a new notion to analyze network training.

At present, there is still a lack of exploration of the memory mechanism of backdoor samples from the frequency domain perspective. Most frequency-based backdoor attack methods make relevant assumptions directly from the existing literature on spectrum exploration or observations of the spectrum, and design triggers based on the assumptions. But they don't understand the specific reasons why triggers generalize and whether the added triggers are the most effective. Wang et al. (2022a) used existing CNN work on spectrum analysis, using YUV and DCT domain transfer to place triggers at medium and high frequencies. Feng et al. based on the idea that the amplitude spectrum can capture low-level distribution and the phase spectrum can capture high-level semantic information. Add triggers to the amplitude spectrum to maintain image semantic information. Zeng et al. (2021) proposed a backdoor attack method to remove ultra-high frequencies to avoid detection based on observations of the spectrum of existing triggers. Hammoud & Ghanem (2021) extracted the k most sensitive frequency components from three channels of the image during training and changed them to different values that carry poisonous information for the attack. These works do not illustrate the real reason why CNN generalizes to its triggers, but we have conducted related explorations. And in order to efficiently implement backdoor attacks on target categories while minimizing damage to the original image energy, we utilize the proposed mechanism to propose a backdoor attack algorithm based on low-frequency semantic information of target categories, which shows excellent attack performance.

## 3 FREQUENCY DOMAIN ANALYSIS

### 3.1 BACKDOOR SAMPLES IN THE FREQUENCY DOMAIN

Taking visible and invisible backdoor attacks as an example, let $D_{train} = (x_i, y_i)_{i=1}^{N}$ represent the training dataset and labels, $\hat{D}_{train} = (\hat{x}_i, y_{target})_{i=1}^{M}$ represent the poisoning training dataset and labels, $f(x_i; \theta)$ represent the classification model parameterized by $\theta$, then $\hat{f}(\hat{x}_i; \hat{\theta})$ represent the model of injecting the backdoor, then the backdoor attack can be expressed as:

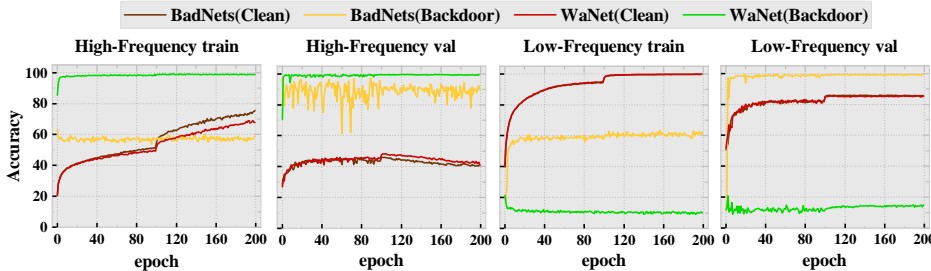

Figure 2: Comparison of models trained using HFC and LFC of images (clean and backdoor images)

$$\hat{f}(x_i; \theta) = y_i \quad , \quad \hat{f}(\hat{x}_i; \hat{\theta}) = y_{target} \tag{1}$$

Same as Wang et al. (2022a), we employ the 2-D discrete cosine transform (DCT) to convert the image to the frequency domain. The spectrum after DCT transformation can concentrate the low-frequency components (shortened as LFC) in the upper left area and high-frequency components (shortened as HFC) of the image in the lower right area. Let $Z_i$ represent the DCT spectrum of $x_i$.

$$Z_i = F(x_i)(k_1, k_2) = \sum_{n_1=0}^{N_1-1} \sum_{n_2=0}^{N_2-1} x(n_1, n_2) c_1(n_1, k_1) c_2(n_2, k_2)$$

$$c_i(n_i, k_i) = \tilde{c}_i(k_i) \cos\left(\frac{\pi(2n_i+1)k_i}{2N_i}\right), \tilde{c}_i(k_i) = \begin{cases} \frac{1}{\sqrt{N_i}} & k_i = 0 \\ \frac{\frac{2}{\sqrt{N_i}}}{\sqrt{N_i}} & k_i \neq 0 \end{cases}, i = 1, 2, \ldots \tag{2}$$

Which transforms a spatial domain image of size $N_1 \times N_2$ into a frequency domain image of the same size. In the formula, $(k_1, k_2)$ represents the index of the spectrum row and column (from top to bottom and left to right, respectively). At the same time, we use the inverse discrete cosine transform (IDCT) $F^{-1}$ to transform the image from the frequency domain to the spatial domain.

CNN will memorize both LFC and HFC during training. Typically, label annotations primarily rely on low-frequency semantic information. As a result, the model will first fit the LFC and gradually fit the HFC to achieve higher training accuracy. However, since the labels of the poisoned samples in the backdoor attacks are specified, the low-frequency semantic vectors are unrelated to the labels. Consequently, the LFC and HFC of images will be treated equally during the training phase. When no trigger is added or the trigger does not work, the model cannot accurately remember a small number of poisoned images. When the trigger becomes effective, it will alter the frequency-domain feature distribution of the original images. The model will memorize the frequency-domain feature distribution introduced by the trigger and establish a connection between the feature distribution and the label.

**Setting.** We set up experiments to verify the inference, using BadNet (Gu et al. (2017)) and IAD (Nguyen & Tran (2020a)) to represent visible backdoor attacks and WaNet (Nguyen & Tran (2020b)) and ISSBA (Li et al. (2021b)) to represent invisible backdoor attacks. To explore the backdoor attack, we use the low-frequency features $x_i^l$ and the high-frequency features $x_i^h$ of clean images and poisoned images to train the model. The training sample generation process is as follows:

$$x_i^l = F^{-1}\left[Z_i \odot m^l\right], \quad \hat{x}_i^l = F^{-1}\left[\hat{Z}_i \odot m^l\right]$$

$$x_i^h = F^{-1}\left[Z_i \odot m^h\right], \quad \hat{x}_i^h = F^{-1}\left[\hat{Z}_i \odot m^h\right] \tag{3}$$

Where $\odot$ is element-wise multiplication, $m^l$ and $m^h$ are the masks for LFC and HFC respectively,

$$m^l = \begin{cases} 1 & , k_1 < \xi \cdot N_1, k_2 < \xi \cdot N_2 \\ 0 & , Otherwise \end{cases}, m^h = \begin{cases} 1 & , k_1 > \xi \cdot N_1, k_2 > \xi \cdot N_2 \\ 0 & , Otherwise \end{cases} \tag{4}$$

**Backdoor attack mechanism.** Comparatively, triggers in visible backdoor attacks generate larger perturbation intensity in the LFC, while triggers in invisible backdoor attacks generate comparatively lower perturbation intensity (Figure 1). Figure 2 illustrates the generalization process of CNN for different frequency components of poisoned and clean images, while other attack methods are

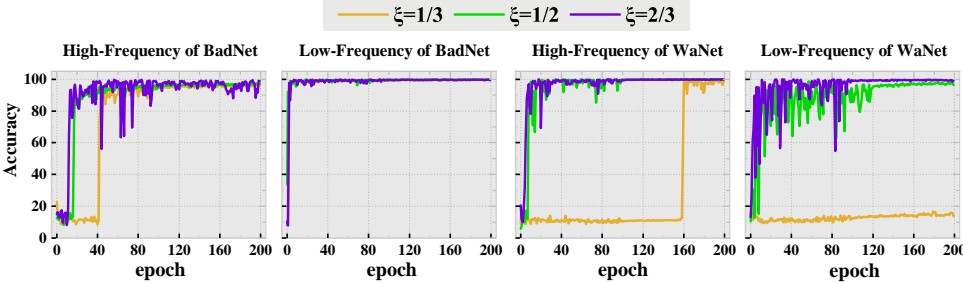

Figure 3: Generalization analysis of CNN perturbed by triggers in various frequency domains ($\xi$ controls trigger perturbation clipping range)

shown in Appendix A.1. When training a model with only high-frequency components, there is no association of low-frequency semantic features with labels. Due to the large number of clean images, CNN can forcibly memorize clean images from HFC, but its whole generalization is limited. Although the number of poisoned images is limited, CNN can quickly memorize the high-frequency feature distribution of trigger changes and produce great generalization. CNN can memorize the low-frequency semantic information of clean samples and generate generalization by solely utilizing LFC to train the model. In the case of poisoned samples, due to the strong low-frequency disturbance generated by the trigger in the visible backdoor attack, CNN can also produce better generalization. Nevertheless, it is difficult to remember the LFC that are changed by the invisible triggers with less low-frequency perturbations.

## 3.2 FREQUENCY DOMAIN ANALYSIS OF BACKDOOR PATTERNS

Figure 2 shows that different attack method triggers activated over different frequency-domain components. Backdoor attacks that are visible seem to be activated throughout both LFC and HFC, whereas invisible backdoor attacks seem to be activated mainly in the mid to high frequency components. As a result, we used multiple visible and invisible backdoor attack methods to further explore how triggers influence on the model generalization capability from various backdoor attacks working in different frequency domains. We generated poisoned samples $\hat{x}_i^{Tl}$ and $\hat{x}_i^{Th}$ by using the low-frequency and high-frequency features produced by the triggers respectively.

$$\hat{x}_i^{Tl} = F^{-1}\left[Z_i \odot \left(1 - m^l\right) + \hat{Z}_i \odot m^l\right], \quad \hat{x}_i^{Th} = F^{-1}\left[Z_i \odot \left(1 - m^h\right) + \hat{Z}_i \odot m^h\right] \quad (5)$$

**Attack effect.** Figure 3 illustrates the influence of the BadNet and WaNet methods in different frequency domain perturbations on attack effectiveness, and the attack effects from other backdoor methods are shown in Appendix A.2. Notably, the triggers in visible backdoor attacks exhibit concentrated disturbances in the LFC, yet the perturbation in other frequency domain components also shows great attack performance. The generalization ability of CNNs on poisoned samples continue to improve with the continuous expansion of the range of intercepted frequency-domain disturbances. The perturbations produced by invisible backdoor attacks at mid and high-frequency can show great generalization, but the model is unable to accurately memorize the perturbations caused by triggers in the LFC. The addition of mid to high-frequency component improves both the generalization capabilities and fitting speed of the model.

It can be seen that although the disturbance of the visible backdoor attack is relatively concentrated in the LFC, the small disturbance generated in the HFC can also show good attack performance. However, the perturbation of invisible backdoor attacks on low-frequency components is unable to complete the generalization of CNN to poisoned samples, showing the difference in generalization of CNN to different perturbation intensities of different frequency domain components. Because the low-frequency component has a strong association with the original labels and occupies a dominant position in most images, we hypothesize that HFC are more susceptible to disturbance than LFC.

To substantiate our hypothesis, we conducted a series of experiments in which we injected fixed random noise as triggers to both LFC and HFC. We used a factor called $\rho$ to adjust the trigger intensity, aiming to assess the critical backdoor perturbation intensity for the LFC and HFC of the samples. The samples with disturbed LFC and HFC are denoted as $\hat{x}_i^{Dl}$ and $\hat{x}_i^{Dh}$ respectively.

$$\hat{x}_i^{Dl} = F^{-1}\left[Z_i + \rho \cdot rand(-1, 1) \odot m^l\right], \quad \hat{x}_i^{Dh} = F^{-1}\left[Z_i + \rho \cdot rand(-1, 1) \odot m^h\right] \quad (6)$$

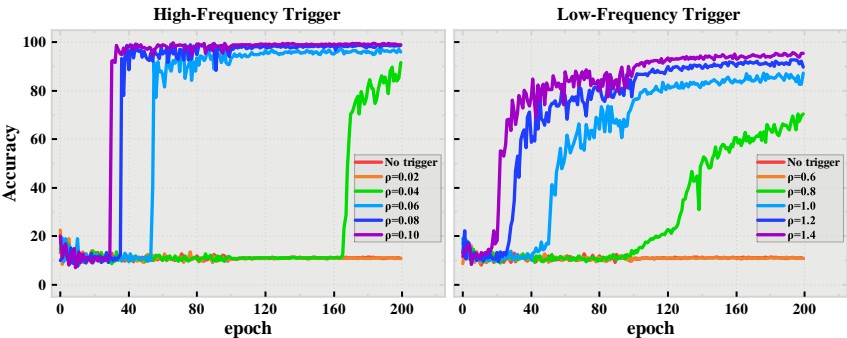

Figure 4: Analysis of backdoor disturbance intensity in HFC and LFC

**Attack intensity analysis.** Figure 4 indicates that achieving a high attack success rate in HFC only requires a trigger intensity of 0.06 for random noise. In contrast, achieving a backdoor attack requires the addition of random noise with an intensity of 0.8 in the LFC, and its accuracy remains lower than that of the HFC. Consequently, high-frequency features are more susceptible to perturbation in the frequency domain, and their attack efficiency is comparatively higher. This experiment also explains why invisible backdoor attacks can achieve backdoor attacks by applying small perturbations in the HFC, whereas invisible backdoor attacks cannot achieve the same through low-frequency perturbations.

## 4 FREQUENCY-BASED BACKDOOR ATTACK STRATEGY

Based on Section 3, we already know that high-frequency perturbations of visible backdoor attacks can achieve backdoor attacks, and smaller perturbation intensity on HFC can make CNN generalize to poisoned samples. In this section, with the aim of reducing the detectability of visible triggers, we propose a strategy to render visible triggers invisible to enhance the stealthiness of visible triggers. We propose a backdoor attack algorithm based on low-frequency semantic information that takes advantage of the CNN's generalization capabilities within different frequency domains of images. This algorithm involves trigger generation and injection in the frequency domain.

### 4.1 RENDERING VISIBLE BACKDOOR ATTACKS INVISIBLE

The visible trigger itself is poorly concealed and is easy to detect during manual inspection. Even small perturbations generated by visible triggers in HFC can lead CNN to generalize poisoned samples. However, small perturbations on HFC are less visible after conversion to the spatial domain. Thus, we propose a universal invisibility strategy for backdoor attacks that renders visible triggers invisible while maintaining their original attack performance.

Given a clean sample $x_i$ and a frequency domain feature mask $m$, a visible trigger is used to generate the poisoned sample $\hat{x}_i$. Upon transforming $x_i$ and $\hat{x}_i$ to the frequency domain, resulting in $Z_i$ and $\hat{Z}_i$ respectively, we generate poisoned samples $\hat{x}_i^{Inv}$ with concealed triggers.

$$\hat{x}_i^{Inv} = F^{-1}\left[Z_i \odot (1-m) + \hat{Z}_i \odot m\right]$$
$$m = \begin{cases} 1 & , & k_1 > \frac{N_1}{2}, k_2 > \frac{N_2}{2} \\ 0 & , & Otherwise \end{cases} \tag{7}$$

**Invisible strategy.** Our proposed strategy can be used for the invisibility of visible triggers. The specific method first uses DCT to transform the poisoned images into the frequency domain, then extracts the disturbance generated by the trigger in the frequency domain, intercepts the mid-high frequency disturbance and adds it to the spectrum. Then it uses IDCT to re-transform images into the space domain to generate the new poisoned samples (Figure 5(a)). The new generated poisoned samples ensure the effectiveness of the original triggers while also attaining their invisibility.

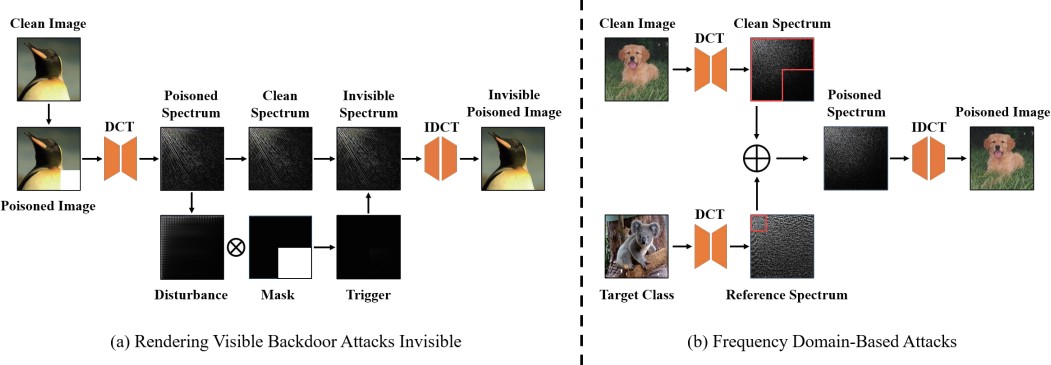

Figure 5: **The pipeline of our attack**. (a) Our strategy for trigger invisibility, which employs medium and high frequency trigger perturbations for hiding. (b) Backdoor attack algorithm based on low-frequency semantic information, injecting target class's low-frequency information into clean image's HFC for backdoor injection.

### 4.2 BACKDOOR ATTACK BASED ON LOW-FREQUENCY SEMANTIC INFORMATION

The LFC of the original image has strong associations with their labels, and small perturbations in the HFC can achieve backdoor attacks. Therefore, we consider using partial low-frequency semantic features of the target class images as raw materials for generating triggers. Furthermore, triggers are injected in the medium to high frequency component to achieve backdoor attacks to ensure maximum invisibility of the triggers.

Given a clean sample $x_i$, a target class sample $x_i^{target}$, and a frequency domain feature mask $m$. We use the DCT to transform $x_i$ and $x_i^{target}$ into the frequency domain to generate $Z_i$ and $Z_i^{target}$ respectively. The trigger intensity factor $\rho$ is used to change the trigger intensity, and an upsampling module $\Phi$ is used to adjust the trigger size to the required dimensions. Let $\hat{x}_i^{Poison}$ represent the poisoned sample.

$$\hat{x}_i^{Poison} = F^{-1} \left[ Z_i + \rho \cdot m \odot \Phi \left[ Z_i^{target} \left( k_1 < \varepsilon \cdot N_1, k_2 < \varepsilon \cdot N_2 \right) \right] \right] \tag{8}$$

**Attack method.** We start by selecting at random a target class image, then transform it into the frequency domain before extracting LFC feature of size $(\varepsilon \cdot N_1, \varepsilon \cdot N_2)$. The feature values are then adjusted to a range of 0 to 1. A linear two-dimensional interpolation method is used to change the size of the low-frequency region into $\left( \frac{N_1}{2}, \frac{N_2}{2} \right)$ as the frequency domain trigger in order to match the size of the trigger dimensions and the perturbation region. The generated frequency domain trigger is applied to the medium to high-frequency components of the clean image's frequency domain, situated in the lower-right region of the DCT spectrum. Finally, we use IDCT to transform it to spatial domain to generate the poisoned sample.

## 5 EXPERIMENT

### 5.1 EXPERIMENTAL SETUP

**Datasets.** We conducted experiments on three widely used datasets: CIFAR-10 (Krizhevsky et al. (2009)), Celeba (Liu et al. (2015)) and MNIST (LeCun et al. (1998)). CIFAR-10 includes ten general categories with a total of 50,000 images. MNIST is a handwritten digit dataset with 60,000 images that includes 10 classifications. Since Celeba is a face dataset, its 40 independent binary attribute annotations make it unsuitable for multi-class classification. As a result, we chose the top three most balanced features, namely Heavy Makeup, Mouth Slightly Open, and Smiling, using the configuration suggested in Salem et al. (2022). These features were used to generate eight groups, totaling 202,599 photos. We also conducted related experiments on Imagenet (Appendix A.5).

**Model.** During the training phase, we used the adam optimizer, initially using a learning rate of 0.01 and decreasing it by a factor of 10 every 100 training steps. We conducted validation using the

Table 1: Validating the strategy for rendering visible triggers invisible

| Dataset → | | CIFAR-10 | | | | MNIST | | | | Celeba | | | |
|---|---|---|---|---|---|---|---|---|---|---|---|---|---|
| | | Original | | Invisible | | Original | | Invisible | | Original | | Invisible | |
| Attack ↓ | Aspect → | BA(%) | ASR(%) | BA(%) | ASR(%) | BA(%) | ASR(%) | BA(%) | ASR(%) | BA(%) | ASR(%) | BA(%) | ASR(%) |
| BadNet | | 94.54 | 99.99 | 94.79 | 99.82 | 99.78 | 100.00 | 99.29 | 100.00 | 79.26 | 100.00 | 77.40 | 99.94 |
| IAD | | 94.50 | 99.33 | 94.09 | 99.98 | 99.54 | 99.56 | 99.68 | 99.93 | 76.77 | 99.87 | 77.22 | 100.00 |

Table 2: Backdoor attack performance

| Model ↓ | Datasets ↓ | Method → | BadNet | IAD | WaNet | ISSBA | Ours $(\rho=0.3, \varepsilon=\frac{1}{4})$ | Ours $(\rho=0.4, \varepsilon=\frac{1}{3})$ | Ours $(\rho=0.5, \varepsilon=\frac{1}{3})$ |
|---|---|---|---|---|---|---|---|---|---|
| ResNet18 | CIFAR-10 | BA(%) | 94.54 | 94.50 | 94.15 | 94.38 | **96.60** | 94.47 | 94.78 |
| | | ASR(%) | 99.99 | 99.33 | 99.55 | 99.99 | **100.00** | **100.00** | **100.00** |
| | MNIST | BA(%) | **99.78** | 99.54 | 99.52 | 99.35 | **99.78** | 99.77 | 99.75 |
| | | ASR(%) | **100.00** | 99.56 | 99.86 | **100.00** | **100.00** | **100.00** | **100.00** |
| | Celeba | BA(%) | 79.26 | 76.77 | 79.42 | 76.06 | 79.05 | **79.71** | 79.25 |
| | | ASR(%) | **100.00** | 99.87 | 99.91 | 99.98 | **100.00** | **100.00** | **100.00** |
| VGG16 | CIFAR-10 | BA(%) | 93.47 | 92.56 | 93.24 | 93.47 | 93.36 | **93.63** | 93.48 |
| | | ASR(%) | **100.00** | 98.71 | 98.90 | 99.98 | **100.00** | **100.00** | **100.00** |
| | MNIST | BA(%) | 99.74 | 99.66 | 99.72 | 99.32 | 99.72 | **99.78** | 99.73 |
| | | ASR(%) | **100.00** | 99.95 | **100.00** | **100.00** | **100.00** | **100.00** | **100.00** |
| | Celeba | BA(%) | 77.39 | 76.17 | 79.68 | 76.06 | 79.05 | **79.71** | 79.25 |
| | | ASR(%) | **100.00** | 99.78 | 99.71 | 99.98 | **100.00** | **100.00** | **100.00** |
| MobileNetV2 | CIFAR-10 | BA(%) | 94.07 | 93.53 | 93.96 | 93.95 | 93.60 | 93.68 | **94.13** |
| | | ASR(%) | 99.99 | 99.25 | 99.70 | 99.98 | 99.98 | **100.00** | **100.00** |
| | MNIST | BA(%) | **99.77** | **99.77** | 99.70 | 99.53 | **99.77** | 99.75 | 99.72 |
| | | ASR(%) | **100.00** | 99.91 | **100.00** | **100.00** | **100.00** | **100.00** | **100.00** |
| | Celeba | BA(%) | 80.01 | 80.08 | 80.03 | 78.24 | 79.74 | 79.87 | **80.20** |
| | | ASR(%) | **100.00** | 99.72 | 99.96 | 99.75 | **100.00** | **100.00** | **100.00** |

ResNet18 (He et al. (2016)), VGG16 (Simonyan & Zisserman (2014)), and MobileNetV2 (Sandler et al. (2018)) models.We also conducted related experiments on vision transform (Appendix A.6).

## 5.2 INVISIBLE ATTACK EXPERIMENT

We used a variety of visible trigger attack algorithms to evaluate the effectiveness of the backdoor hiding effect on the ResNet18 model. Appendix A.3 contains the results of the backdoor hiding effect. The attack performance after rendering visible triggers invisible is shown in Table 1. We used Attack Success Rate (ASR) and Benign Accuracy (BA) to assess the efficacy of various attacks. After achieving the invisibility of visible triggers, BadNet maintained ASR of 99.82%, 100%, and 99.94% on the CIFAR-10, MNIST, and Celeba datasets respectively. Only the CIFAR-10 and Celeba datasets had a small decline in ASR of 0.17% and 0.06% respectively. On the other hand, their BA increased by 0.25% and 0.14% respectively. After the visible trigger is concealed, IAD exhibits an enhanced ASR across all datasets. This indicates the success of our backdoor invisible strategy.

## 5.3 FREQUENCY DOMAIN-BASED ATTACK EXPERIMENTS

We evaluated the impact of different $\rho$ and $\varepsilon$ on BA and ASR, as well as the degree of the trigger invisibility. Appendix A.4 contains the results, which show that $\rho$ is a relatively important factor influencing both BA and ASR. We selected three $\rho$ and $\varepsilon$ combinations, which demonstrated promising overall performance, and conducted comparison analysis against various backdoor attack algorithms across multiple datasets and models. The results show that our algorithm achieves an ASR of nearly 100% on various datasets and models, with an average BA improvement of 0.2%. Notably, on the CIFAR-10 dataset, the BA of the ResNet18 has increased by 2.06%.

## 5.4 DEFENSE EXPERIMENTS

We used defense methods and network visualization tools to verify the effectiveness of the proposed backdoor attack model. We also conducted experiments on other defense methods.(Appendix A.7)

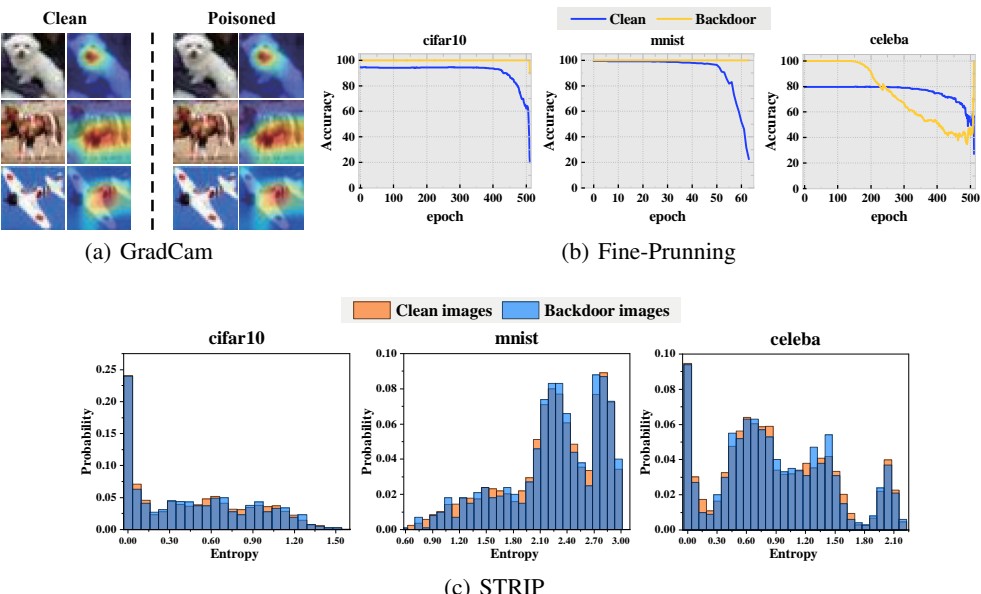

Figure 6: Backdoor defense experiment of our attack method

Fine-Pruning (Liu et al. (2018a)) is a model defense method based on neuron analysis. The method analyzes neuronal responses to a set of clean images and detects and gradually prunes dormant neurons to mitigate the impact of backdoors. Figure 6(b) shows the pruning effect of Fine-Pruning applied to our model. The accuracy of clean data does not exceed that of the attacked data across all datasets, making backdoor mitigation nearly impossible.

STRIP (Gao et al. (2019)) is a backdoor defense method based on input filtering. In this approach, known input images are perturbed by introducing clean images of different categories, and the change in predictive entropy between the mixed and original images is compared. Figure 6(c) demonstrates that the entropy distribution of our poisoned model closely resembles that of the clean model, indicating that the poisoned model trained using our poisoned samples exhibits behavior similar to a clean model.

GradCam (Selvaraju et al. (2017)) is a network visualization tool that helps in examining network behavior. Since our attack method is injected from high-frequency components, the perturbation is concentrated in the image details and therefore cannot be detected by it. Figure 6(a) shows that the visual heatmaps of the clean model and the poisoned model are basically consistent.

## 6 CONCLUSION

In this paper, we analyzed the impact of triggers on the generalization of CNN from the perspective of frequency domain, and proved that triggers change the frequency domain feature distribution of images to make CNN generalize. We investigated how changes in trigger-induced alterations across different frequency components contribute to generalization. We also proved that high-frequency components are more easily perturbed than lower-frequency components and have higher attack efficiency. Based on these findings, we proposed a simple yet effective universal strategy to achieve invisibility of visible triggers, ensuring minimum losses in both BA and ASR. Furthermore, we proposed a backdoor attack algorithm based on low-frequency semantic information that achieved significant results and was able to circumvent many defenses.

Our interpretation of CNN memory poisoning samples helps solve other problems. For example, by maintaining the same frequency domain feature distribution, we can consider setting invisible triggers in the training phase, and setting physically realizable triggers in the inference phase. Our discussion of backdoor defense mechanisms is still a bit inadequate, and we will improve this part in the next work. We hope this research will inspire more in-depth discussions about the characteristics of backdoors, thereby promoting the further development of backdoor attacks and defenses.

ACKNOWLEDGMENTS

The research is supported by the New Liberal Arts Research and Reform Practice Project of Ministry of Education of China (No. 2021060049), the Postgraduate Education and Teaching Reform Research Project of Shandong (No. SDYJG21185), the Key Project of Undergraduate Teaching Reform Research of Shandong (No. Z2021323), the Humanity and Social Science Research Project of Ministry of Education of China (No. 20YJAZH069, No. 20YJC740062), and the Social Science Foundation of Shanghai (No. 2019BYY028).

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

# A APPENDIX

We evaluate multiple backdoor attack algorithms, including visible backdoor attacks and invisible backdoor attacks. Experiments on these algorithms can prove our conclusions. Due to space reasons, we list representative results in the main text.

## A.1 ANALYSIS OF IMAGE'S HIGH AND LOW-FREQUENCY COMPONENTS

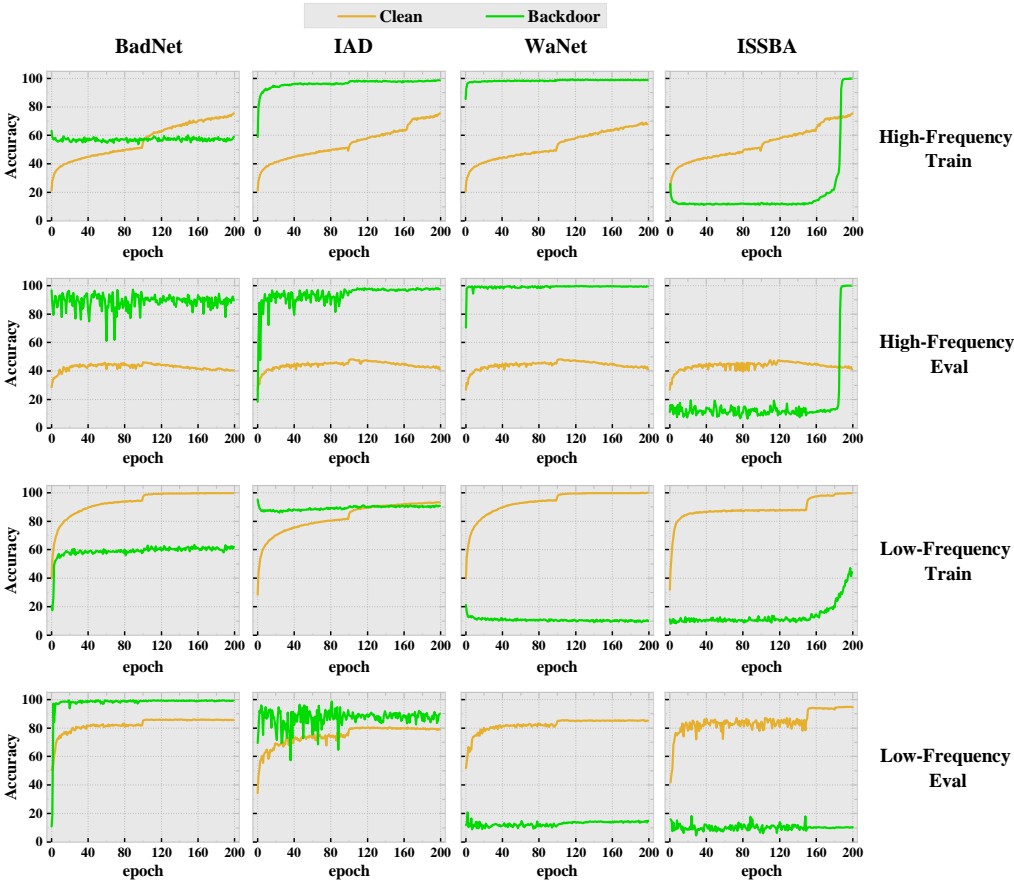

Figure 7: **CNN generalization analysis of different frequency domain images** (including clean images and backdoor images). Each column in the figure corresponds to a backdoor attack method, a total of 4 columns, and each row corresponds to the model generalization process of an attack method on high or low frequency, training set or validation set.

We set up experimental verification inferences in this paper, using BadNet and IAD as visible backdoor attacks and WaNet and ISSBA as invisible backdoor attacks for exploration. For training the model, we extract low-frequency features $x_i^l$ and high-frequency features $x_i^h$ from both clean and poisoned images. The generation of training samples is shown in equations 3 and 4. Among these, we set $\xi$ to $\frac{1}{3}$.

In Figure 7, we present a visual representation of the influence of high and low-frequency information extracted from both clean and poisoned images on the model's generalization. This confirms that the generalization ability of CNN on poisoned samples is achieved through memorizing the frequency domain features after injected by triggers, prompting the model to exhibit generalization tendencies in the presence of a limited number of poisoned images, thus enabling the successful execution of backdoor attacks.

## A.2 ANALYSIS OF HIGH AND LOW-FREQUENCY PERTURBATIONS IN TRIGGERS

Figure 8: **CNN generalization analysis of different trigger perturbations in different frequency domains** ($\xi$ control trigger perturbation limiting range). Each column in the figure corresponds to a backdoor attack method, a total of four columns, and each row corresponds to a model generalization process of attack method on high or low frequency, training or verification sets.

We extended our investigation with BadNet and IAD as visible backdoor attacks and WaNet and ISSBA as invisible backdoor attacks to explore the effects of perturbations in different frequency domains on model generalization. To accomplish this, we generated poisoned samples $\hat{x}_i^{Tl}$ and $\hat{x}_i^{Th}$ using only low-frequency and high-frequency features produced by triggers, respectively.

Figure 8 illustrates the impact of perturbations in various frequency domains on model generalization performance across different attack methods. The parameter $\xi$ controls the range of low and high-frequency component truncation in this case, thus enabling the extraction of distinct frequency domain features. The generation of training samples is shown in equations 4 and 5.

## A.3 RENDERING VISIBLE TRIGGERS INVISIBLE

In this section, we evaluated the invisibility effectiveness of our invisible algorithm for visible triggers. To achieve trigger invisibility, we extracted effective perturbations from the original trigger in the mid-to-high frequency range while suppressing low-frequency disturbances. Figure 9 shows the invisibility effect. The visible trigger in the spatial domain becomes invisible to humans, showing an excellent invisibility effect. Furthermore, even when invisibly hidden, the algorithm retains its benign accuracy (BA) and attack success rate (ASR), indicating the efficacy of our invisibility technique.

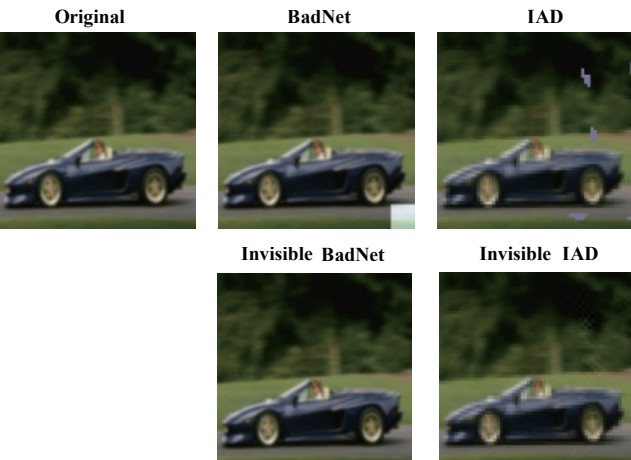

Figure 9: **Render the visible trigger invisible.** The first row is the image of the original sample and the trigger with visible backdoor attack added, and the second row is the image processed by our strategy.

## A.4 FREQUENCY DOMAIN TRIGGER ANALYSIS

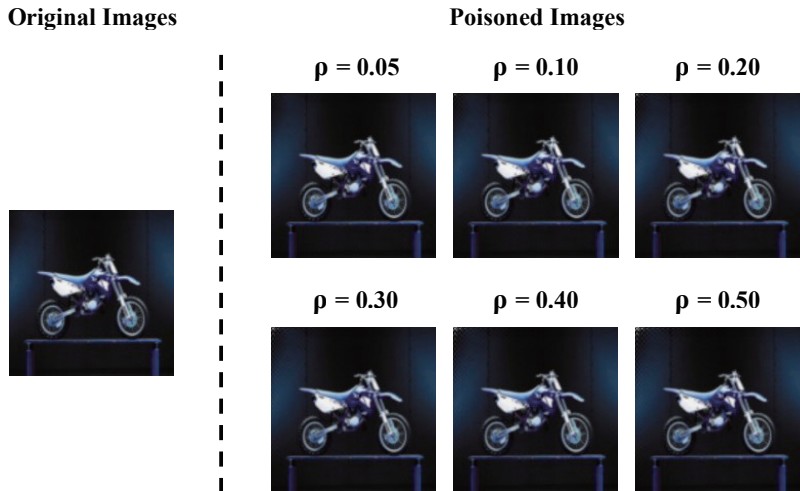

Figure 10: The poisoned image is generated by our backdoor attack algorithm based on low-frequency semantic information. The intensity factor $\rho$ governs the range of values for the frequency domain trigger, set to $\frac{1}{3}$ for $\varepsilon$.

We evaluated the impact of varying $\rho$ and $\varepsilon$ on BA, ASR, and trigger invisibility on the CIFAR-10 dataset. Table 3 shows the results of the backdoor attacks, and Figure 10 shows the triggers' invisibility. Given that $\varepsilon$ has only a small impact on trigger visibility, we merely showed trigger visibility for different $\rho$ values. We can observe that when the trigger strength is excessive, partial artifacts may appear in the upper-left corner of the spatial domain due to the nature of the DCT transformation. At $\rho = 0.40$, careful scrutiny is required to detect the image perturbation, while at $\rho = 0.50$, the changes generated by the trigger in the upper-left corner of the image become noticeable to the human. However, high-intensity triggers exhibit significant attack effectiveness and can bypass multiple defenses, while even lower-intensity triggers continue to demonstrate strong attack efficacy and performance. Therefore, our attack method remains successful.

At the same time, we also selected three combinations of $\rho$ and $\varepsilon$ with good comprehensive effects to verify the influence of different poisoning rates on attack methods (Table 4). The ResNet18 model

Table 3: Comparison of intensity factor $\rho$ and low-frequency semantically triggered region selection factor $\varepsilon$

| factor $\rho \downarrow$ factor $\varepsilon \rightarrow$ | $\varepsilon = \frac{1}{8}$ | | $\varepsilon = \frac{1}{4}$ | | $\varepsilon = \frac{1}{3}$ | | $\varepsilon = \frac{1}{2}$ | |
|---|---|---|---|---|---|---|---|---|
| | BA | ASR | BA | ASR | BA | ASR | BA | ASR |
| $\rho$=0.10 | 82.50 | 98.21 | 82.09 | 99.14 | 82.42 | 99.36 | 81.15 | 99.15 |
| $\rho$=0.20 | 94.52 | 99.90 | 94.44 | 99.97 | 94.64 | 99.97 | 94.46 | 99.97 |
| $\rho$=0.30 | 94.48 | 99.99 | 96.60 | 100.00 | 94.61 | 99.95 | 94.85 | 100.00 |
| $\rho$=0.40 | 94.69 | 99.97 | 94.46 | 100.00 | 94.47 | 100.00 | 94.70 | 100.00 |
| $\rho$=0.50 | 94.58 | 100.00 | 94.67 | 100.00 | 94.78 | 100.00 | 94.67 | 100.00 |

and CIFAR-10 dataset were used in the experiment. Our attack method can basically achieve a 100% attack success rate when the poisoning rate is from 0.02 to 0.10, and the variation range of the benign accuracy rate is around 1%. This shows that our method can still achieve high performance under the condition of low poisoning rate.

Table 4: Performance of our attack method at different poisoning rates

| Rates $\downarrow$ Model $\rightarrow$ | $\rho = 0.3, \varepsilon = \frac{1}{4}$ | | $\rho = 0.4, \varepsilon = \frac{1}{3}$ | | $\rho = 0.5, \varepsilon = \frac{1}{3}$ | |
|---|---|---|---|---|---|---|
| | BA | ASR | BA | ASR | BA | ASR |
| 0.02 | 94.99 | 99.96 | 93.83 | 100.00 | 94.44 | 100.00 |
| 0.04 | 94.78 | 99.99 | 94.27 | 100.00 | 94.52 | 100.00 |
| 0.06 | 95.69 | 100.00 | 94.25 | 100.00 | 94.50 | 100.00 |
| 0.08 | 95.58 | 100.00 | 94.41 | 100.00 | 94.56 | 100.00 |
| 0.10 | 96.60 | 100.00 | 94.47 | 100.00 | 94.78 | 100.00 |

## A.5 VALIDATION OF INVISIBLE STRATEGIES AND ATTACK ALGORITHMS ON IMAGENET

One of the benchmark datasets for image classification, ImageNet(Deng et al. (2009)), consists of 1000 classes. Due to high computational and memory costs, the entire dataset is rarely utilized in existing backdoor learning efforts(Wu et al. (2022)). We used Tiny-ImageNet(Le & Yang (2015)), which is currently widely used in the field of backdoor attacks. The dataset contains a subset of 200 classes, with 100,000 images for training (500 images per class) and 10,000 images for testing (50 images per class). We used the pre-resnet18 model to conduct relevant experiments. The poisoning rate is set to 10%.

As can be seen from Table 5, our backdoor attack algorithm is still efficient on the Tiny-ImageNet dataset. Some other algorithms have low BA or low ASR on this data set, but our attack algorithm still maintains an attack rate of 100.00%, and is also better at classifying cleandata. Table 6 shows the attack effect of our invisible algorithm on BadNet. This shows that our algorithm still works on the ImageNet dataset, ensuring the original performance of the algorithm.

## A.6 VALIDATION OF INVISIBLE STRATEGIES AND ATTACK ALGORITHMS ON VISION TRANSFORMER

We evaluate our algorithm on a vit-base model using the cifar10 dataset. The input patch size of the vit-base model is $14 \times 14$, and the poisoning rate is set to 10%. It can be seen from the experimental results (Table 7) that our attack method is still applicable in vision transformer. And compared with

Table 5: Backdoor attacks on ImageNet

| Method ↓       Aspect → | BA | ASR |
|---|---|---|
| LIRA Doan et al. (2021) | 0.05 | 99.85 |
| BPP Wang et al. (2022b) | 58.14 | **100.00** |
| Blind Bagdasaryan & Shmatikov (2021) | 1.97 | 45.21 |
| TrojanNN Liu et al. (2018b) | 55.89 | 99.98 |
| Ours    ($\rho = 0.4, \varepsilon = \frac{1}{3}$) | **59.11** | **100.00** |

Table 6: The implementation of invisible strategies on ImageNet

| Method ↓    Aspect → | Origin | | Invisible | |
|---|---|---|---|---|
|  | BA | ASR | BA | ASR |
| BadNet | 65.31 | 97.17 | 61.18 | 99.16 |
| IAD | 57.58 | 99.67 | 57.99 | 99.98 |

other backdoor attack methods, our attack algorithm achieves high BA and ASR. Table 8 shows the effectiveness of our stealth strategy on badnet. After using our stealth strategy, we not only achieve the stealth of visible triggers, but also ensure the original attack performance, and even achieve a certain amount of improvement in ASR.

Table 7: Backdoor attack on vision transformer

| Method ↓        Aspect → | BA | ASR |
|---|---|---|
| BadNet Gu et al. (2017) | 94.78 | 93.77 |
| Blended Chen et al. (2017) | 96.54 | 99.68 |
| LIRA Bagdasaryan & Shmatikov (2021) | 17.52 | 87.87 |
| TrojanNN Liu et al. (2018b) | 96.46 | 99.98 |
| Ours($\rho = 0.4, \varepsilon = \frac{1}{3}$) | **98.62** | **100.00** |

## A.7 EXPERIMENTATION WITH OTHER BACKDOOR DEFENSE METHODS

Different from the method used in section 5.4, we use other defense methods to defend our attack algorithm to demonstrate the effectiveness of our algorithm.

DBD(Huang et al. (2021)) decouples the original end-to-end training process into three stages, and using strong data augmentation involved in self-supervised learning destroys the triggering pattern, making it unlearnable in representation learning. The decoupling process further disconnects trigger modes and target tags. Implemented that even if the model is trained on a toxic dataset, the hidden backdoor cannot be successfully created. Table 9 shows that this defense is not effective against our attack method, and our attack algorithm can still achieve high ASR.

NC(Wang et al. (2019)) were the first to propose a robust and general detection and mitigation system for DNN backdoor attacks. Their technology can identify multiple existing backdoors and reconstruct possible trigger points. Use the model to reversely infer the position and shape of the trigger, and then determine whether there is a backdoor; secondly, use the similar impact of the reverse trigger and the actual trigger on neuron activation to mitigate backdoor attacks. Many existing backdoor attacks will lose the performance of backdoor attacks after being mitigated by NC, but the mitigating effect of NC on our method is not very great. As can be seen from Table 9, the ASR of our backdoor attack algorithm can still reach 91.80%

Table 8: The implementation of invisible strategies on vision transformer

| Method ↓ | Aspect → | Origin | | Invisible | |
|---|---|---|---|---|---|
| | | BA | ASR | BA | ASR |
| BadNet | | 94.78 | 93.77 | 94.77 | 99.12 |
| IAD | | 98.37 | 99.26 | 98.25 | 99.23 |

CLP(Zheng et al. (2022)) introduced a new concept, namely the Lipschitz constant for each channel's mapping from the input image to the output. Empirical evidence was then used to show that there is a strong correlation between the upper bound of CLC (UCLC) and changes in channel activation that trigger activation. Since UCLC can be calculated directly from the weight matrix, they can detect potential backdoor channels in a data-free manner and perform simple pruning of infected DNNs to repair the model. Table 9 shows the defense results of CLC against our method. The results show that CLC is ineffective. Although the ASR of our attack algorithm has slightly decreased under his defense, it is still effective.

Table 9: Mitigation results of backdoor defense methods on our algorithm

| Attack ↓ Defense → | DBD | | NC | | CLP | |
|---|---|---|---|---|---|---|
| | BA(%) | ASR(%) | BA(%) | ASR(%) | BA(%) | ASR(%) |
| SIG Barni et al. (2019) | 60.67 | 100.00 | 84.48 | 98.27 | 83.80 | 98.91 |
| Blended Chen et al. (2017) | 69.91 | 99.68 | 93.47 | 99.92 | 91.32 | 99.74 |
| WaNet Nguyen & Tran (2020b) | 80.90 | 6.61 | 91.80 | 7.53 | 81.91 | 78.42 |
| ISSBA Li et al. (2021a) | 63.50 | 99.51 | 90.99 | 0.58 | 91.38 | 68.13 |
| Ours($\rho = 0.4, \varepsilon = \frac{1}{3}$) | 70.29 | 100.00 | 94.26 | 91.80 | 94.55 | 98.41 |

## A.8 DISCUSSION WITH OTHER RELATED WORK

**Discussion with Zeng et al. (2021).** Zeng et al. (2021) studied the spectrum of poisoned images and showed that most triggers will produce artifacts in the high-frequency components of large-sized input images, and explained the reasons for this phenomenon accordingly. That is, the high-frequency artifacts of Local Patching mainly originate from the high-frequency components of the trigger mode itself. High-frequency artifacts of Large-Size or Global Patching originate from the trigger pattern itself or the method of inserting the trigger. GAN-Generated Backdoor Data originates from the upsampling operation adopted by gan. We explained the mechanism of CNN memory triggers from the perspective of frequency domain, indicating that CNN can memorize specific frequency domain feature distribution and establish the association between feature distribution and labels. Therefore, backdoor attacks can be achieved even if the trigger factors are diverse or not fixed. We used this mechanism to explore the impact of perturbations produced by visible and invisible triggers in different frequency domain components on the generalization ability of CNN, as well as the differences in the generalization ability of CNN to different perturbation intensities imposed on different frequency domain components. It is also proven that high-frequency components are more sensitive to perturbations relative to low-frequency components.

**Discussion with Feng et al..** Feng et al. uses FFT to transform the image from the spatial domain to the frequency domain, and then linearly adds the low-frequency component of the clean image and the low-frequency component of the target image through a certain ratio as the low-frequency component of the poisoned image, thereby achieving frequency injection of the trigger. Literature [3] uses FFT to transform the image from the spatial domain to the frequency domain, and then linearly adds the low-frequency component of the clean image and the target image through a certain ratio as the low-frequency component of the poisoned image, thereby achieving frequency injection of the trigger. And their training process introduced noise images similar to those in the literature [4] to enhance the uniqueness of the trigger. However, our frequency-based backdoor attack method

selects part of the low-frequency components of the target category image as the raw material of the trigger. The poisoned image is then generated by adding triggers to the mid- and high-frequency components of the clean image through operations such as upsampling. And in order to improve the generalization of CNN to poisoned images whose label is the target category, we randomly select the low-frequency components of the image from the target category as raw materials when generating each poisoned image, and do not use noise images.

**Discussion with Yin et al. (2019).** The work of Yin et al. (2019) mainly analyzes the trade-offs between two data enhancement strategies, Gaussian enhancement and adversarial training, for different corruption types from the perspective of the frequency domain. They found that the augmentation strategy made the model tend to exploit low-frequency information in the input. This low-frequency bias reduces susceptibility to high-frequency damage and improves robustness to high-frequency damage while degrading performance against low-frequency damage. They also demonstrated that AutoAugment, a data augmentation, can play a key role in mitigating robustness issues. In our work, we mainly explored the mechanism by which CNN generates generalization for backdoor samples, and analyzed the role of existing triggers in frequency domain perturbations. The vulnerability of high versus low frequency components is also explored. So our theory is different.

**Discussion with Wang et al. (2020).** Our work is inspired by Wang et al. (2020) and attempts to explore the generalization of CNN to backdoor samples from the frequency domain perspective. The contribution of Wang et al. (2020) is that they first noticed the ability of CNN in capturing high-frequency components of images. And from a data perspective, it is revealed that CNN achieves a trade-off between accuracy and robustness by utilizing the high-frequency components of images. They proposed that in generally labeled samples, the model first extracts low-frequency information and then gradually extracts high-frequency information to achieve higher training accuracy. In samples with chaotic labels, the model treats low-frequency and high-frequency information equally, which means that the model begins to choose to memorize the data. This part of the theory is also a hypothesis for our work. They also further explored the impact of training techniques on training results from a frequency perspective. For example, they found that a larger batch size has smaller changes in high-frequency information, narrowing the generalization gap, so it will lead to a smaller gap between training accuracy and test accuracy. For adversarial learning, they found that adversarial learning has a smaller generalization gap, making the system convolution kernel smoother, and making a trade-off between accuracy and robustness. The discussion on the generalizability of backdoor samples in our theory is based on their theory, but the subsequent analysis of the perturbation of backdoor samples in the frequency domain was proposed and proved by us.

