# OpenReview forum: "Rethinking CNN’s Generalization to Backdoor Attack from Frequency Domain"
_ICLR.cc/2024/Conference — ICLR 2024 poster_

### Official Review · Reviewer_6jrw · 2023-10-26

**Soundness:** 2 fair
**Presentation:** 3 good
**Contribution:** 3 good
**Rating:** 6
**Confidence:** 3

**Summary:**

This paper focuses on the mechanism of CNN memorize poisoned samples in the frequency domain, and finds that CNN generate generalization to poisoned samples by memorizing the frequency domain distribution of trigger changes. It proposes a universal invisible strategy for visible triggers, which can achieve trigger invisibility while maintaining raw attack performance. It also designs a novel frequency domain backdoor attack method based on low-frequency semantic information. The main contributions of this paper are:
- It explores the mechanism of CNN memorization for poisoned images from a frequency domain perspective, investigating the generalization of CNN with respect to perturbations in different frequency domains.
- It explores the generalization of CNN for visible and invisible triggers, demonstrating that high-frequency features are more susceptible to perturbations than low-frequency features.
- It proposes a generalized strategy for rendering visible backdoor attacks invisible while maintaining algorithmic performance.
- It proposes a backdoor attack algorithm based on low-frequency semantic information for target classes, achieving high success rates across diverse datasets and models.

**Strengths:**

The main strengths of the paper are:
- The paper is well-organized and clearly written, which is easy to follow.
- This paper provides novel perspectives for backdoor attacks on CNN.
- Experimental results are promising and can validate the effectiveness of the proposed method.

**Weaknesses:**

The weaknesses of the paper are:
- The motivation of this paper is unclear. The authors mention some related work about CNN and backdoor attacks, and they put forward their findings and the proposed methods. However, the necessity and urgency of the proposed method are still unclear. The authors should further clarify the motivation of the paper.
- The dataset used in the paper is relatively small, which is not sufficient to reflect the generalization of the experiment. The author should conduct experimental observations on more generalized datasets such as ImageNet.

------------------------------------After Rebuttal-----------------------------------------

I thank the authors for their response. The response addressed most of my concerns. After reading other reviewers' comments, I think this paper is marginally above the acceptance threshold, thus I keep my score.

**Questions:**

- Q1. The motivation of this paper is unclear. The authors mention some related work about CNN and backdoor attacks, and they put forward their findings and the proposed methods. However, the necessity and urgency of the proposed method are still unclear. The authors should further clarify the motivation of the paper.
- Q2. The dataset used in the paper is relatively small, which is not sufficient to reflect the generalization of the experiment. The author should conduct experimental observations on more generalized datasets such as ImageNet.

---

> ### Author Response · Authors · 2023-11-18
> **Response to Reviewer 6jrw (Part 1)**
>
> Thank you very much for your time and insightful comments. We are encouraged that our work is recognized. We hope the following new clarifications and results can address your concerns. We are happy to answer more questions and conduct more experiments if needed.
>
> Q1: The motivation of this paper is unclear. The authors mention some related work about CNN and backdoor attacks, and they put forward their findings and the proposed methods. However, the necessity and urgency of the proposed method are still unclear. The authors should further clarify the motivation of the paper.
>
> A1: Most current studies only study the impact of changing specific trigger features in the spatial domain on model generalization. The limitations of the spatial domain make it difficult for us to understand how CNN generalizes to poisoned samples. Frequency domain analysis is a good idea for analyzing the training process of neural networks. [1] provides a good idea for understanding the problem of how well deep neural networks can generalize. At the same time, [2] explained the high-frequency components from the perspective of the frequency domain to help CNN generalize. This study has given us some inspiration. Let us try to explain the mechanism of CNN's generalization on backdoor samples from the perspective of frequency domain. So we took visible and invisible triggers as examples to explore CNN's memory mechanism for poisoned images from the frequency domain perspective. We prove that successful triggers change the frequency domain distribution of samples, and CNN can generalize to a small number of poisoned samples by memorizing the frequency domain distribution changed by triggers. Our experiments also prove that the frequency domain distribution changed by the trigger is not always effective. For some triggers, only a part of it is effective. For some triggers, only a part of the perturbation can be used to achieve an efficient attack. Moreover, different frequency domain distributions of samples have different vulnerabilities to disturbances, and high-frequency components are more susceptible to disturbance than low-frequency components. In addition, after CNN establishes the association between feature distribution and labels, even if the triggering factors are diverse or unstable, backdoor attacks can be implemented as long as the frequency domain distribution is the same.
>
> Visible backdoor attacks exhibit a higher success rate, but their applicability is limited in scenarios that demand a higher level of concealment due to the visibility of their triggers. Therefore, combined with our theory, we extract the high-frequency component generated by the trigger that has less perturbation on the spectrum energy, and use the perturbation of the high-frequency component to inject a backdoor into the model. The spectral transformation of high-frequency perturbations into the spatial domain achieves the invisibility of triggers, thereby ensuring the efficiency of visible triggers in attacks. This expands the scenarios in which visible triggers can be effectively employed. Our proposed backdoor attack method, based on frequency-domain principles, capitalizes on the theoretical framework we introduced. This method harnesses the CNN's memory mechanisms for backdoor samples more effectively, enabling a highly efficient implementation of backdoor attacks. As demonstrated in Section 5, the attack accuracy of this method reaches 100% across multiple datasets and models.
>
> We added the necessity of our proposed method in section 1 of the paper.
>
> [1] Zhang Y, Xu Z Q J, Luo T, et al. Explicitizing an implicit bias of the frequency principle in two-layer neural networks[J]. arXiv preprint arXiv:1905.10264, 2019.
>
> [2] Haohan Wang, Xindi Wu, Zeyi Huang, and Eric P Xing. High-frequency component helps explain the generalization of convolutional neural networks. In Proceedings of the IEEE/CVF conference on computer vision and pattern recognition, pp. 8684–8694, 2020.

---

> > ### Author Response · Authors · 2023-11-18
> > **Response to Reviewer 6jrw (Part 2)**
> >
> > Q2: The dataset used in the paper is relatively small, which is not sufficient to reflect the generalization of the experiment. The author should conduct experimental observations on more generalized datasets such as ImageNet.
> >
> > A2：Thank you for pointing out our experimental shortcomings. To make up for the shortcomings, we conducted experiments on our related algorithms on ImageNet. However, due to GPU resource and time constraints, we currently use Tiny-ImageNet, which is widely used in the field of backdoor attacks. The dataset contains a subset of 200 classes, with 100,000 images for training (500 images per class) and 10,000 images for testing (50 images per class) [3]. We verified the backdoor attack (Table 1) and the stealth strategy (Table 2) with parameters set to (ρ = 0.4, ε = 1/3). We can see that our backdoor attack method and invisible strategy can still achieve good attack performance on ImageNet. The results obtained on ImageNet have been placed in Appendix A.5.
> >
> > Table 1: Backdoor attack on ImageNet
> >
> > |                                        | **BA** | **ASR** |
> > |----------------------------------------|--------|---------|
> > | LIRA                       | 0.05   | 99.85   |
> > | BPP                         | 58.14  | **100.00** |
> > | Blind                     | 1.97   | 45.21   |
> > | TrojanNN                   | 55.89  | 99.98   |
> > | Ours($\rho=0.4$,$\varepsilon=\frac{1}{3}$) | **59.11** | **100.00** |
> >
> > Table 2: The Implementation of Invisible Strategies on ImageNet
> >
> > |         | **Origin BA** | **Origin ASR** | **Invisible BA** | **Invisible ASR** |
> > |---------|---------------|----------------|------------------|-------------------|
> > | BadNet  | 65.31         | 91.17          | 61.18            | 99.16             |
> >
> > [3] Wu B, Chen H, Zhang M, et al. Backdoorbench: A comprehensive benchmark of backdoor learning[J]. Advances in Neural Information Processing Systems, 2022, 35: 10546-10559.

---

> > > ### Comment · Reviewer_6jrw · 2023-11-23
> > > **Response to Authors**
> > >
> > > I thank the authors for their response. The response addressed most of my concerns. After reading other reviewers' comments, I think this paper is marginally above the acceptance threshold, thus I keep my score.

---

> > > > ### Author Response · Authors · 2023-11-23
> > > > **Sincere thanks to reviewer 6jrw**
> > > >
> > > > We are pleased to have addressed your question well and thank you for reviewing our paper.

---

### Official Review · Reviewer_2gr1 · 2023-10-27

**Soundness:** 3 good
**Presentation:** 2 fair
**Contribution:** 2 fair
**Rating:** 5
**Confidence:** 3

**Summary:**

This paper investigate CNN's generalization to backdoor attack in the frequency domain. Based on the fact that high frequencies are more easily perturbed and have higher attack efficiency. They designed an algorithm to convert visible triggers into invisible triggers and also a backdoor attack in the frequency domain.

**Strengths:**

1. This paper is well organized.

2. The both proposed algorithms are well-motivated.

**Weaknesses:**

1. The authors claimed that "backdoor attack and defense from a frequency domain perspective is
still insufficient", but IMHO, their literature review is insufficient. Recently, there are many researches about backdoor attacks in the frequency domain. The authors only mentioned one work [1], please discuss more related works such as [2,3,4]. By the way, adversarial attack (aka evasion attack) is closely related to backdoor attack, and there are even more adversarial attacks [5,6] in the frequency domain, and I noticed that the proposed algorithms shown in Figure 5(b) share some similarities with f-mixup in [6].

2. [7,8] has already provided comprehensive analyses about CNN's generalization in the frequency domain, I think most conclusions reached in Section 3 may also be derived from the previous work. In other words, this paper does not provide enough new insights as the authors claimed.

3. The proposed attack is motivated by frequency domain analysis for CNNs , but ViT has different bias in the frequency domain [9], which means that the proposed method may not generalize to ViT. IMHO, this is a great limitation.

4. The authors conducted experiments on MNIST, CIFAR-10 and Celeba, it is necessary to validate the proposed algorithms on ImageNet. In my experience, CNNs trained on ImageNet is far less sensitive to high frequencies than CNNs trained on the above datasets.

Overall, although I think this work is interesting, the current manuscript seems to not achieve the bar of ICLR in 2024.


Reference

[1]  An invisible black-box backdoor attack through frequency domain, ECCV 2022.

[2] FIBA: Frequency-Injection based Backdoor Attack in Medical Image Analysis, CVPR 2022.

[3] Check Your Other Door! Creating Backdoor Attacks in the Frequency Domain, BMVC 2022.

[4] Rethinking the Backdoor Attacks’ Triggers: A Frequency Perspective, ICCV 2021.

[5] Surfree: a fast surrogate-free black-box attack, CVPR 2021.

[6] Decision-based adversarial attack with frequency mixup, IEEE TIFS 2022.

[7] A Fourier perspective on model robustness in computer vision, NeurIPS 2019.

[8] High-frequency component helps explain the generalization of convolutional neural networks, CVPR 2020.

[9] How Do Vision Transformers Work? ICLR 2022

------------------------------------After Rebuttal-----------------------------------------

The authors' rebuttal has addressed most of my concerns. However, the insufficient literature review still heavily weakens the motivation and novelty of this work. Specifically, the authors claimed that "the research on backdoor attack and defense from a frequency domain perspective is still insufficient", but actually, there exist many works in this direction. Besides, although the proposed algorithms shown in Figure 5(b) differs from f-mixup in some details, I still think the high-level ideas are similar, since both of inject misleading features into middle-and-high frequency domain. Therefore, although I have raised my score to 5, I think it is okay to reject this paper.

**Questions:**

Please see weakness.

---

> ### Author Response · Authors · 2023-11-18
> **Response to Reviewer 2gr1 (Part 1)**
>
> Thank you very much for your time and insightful comments. We are encouraged that our work is recognized. We hope the following new clarifications and results can address your concerns. We are happy to answer more questions and conduct more experiments if needed.
>
> Q1: Insufficient review and some similarities with f-mixup in literature [1]
>
> A1: Thanks for pointing this out. In fact, we previously included a review of frequency domain backdoor attacks, but due to space reasons we had to reduce the introduction to this aspect, which we admit is insufficient. We have now supplemented the related work with a review of other backdoor attack methods (section 2.2).
>
> Regarding the work you mentioned in the field of adversarial attacks [1], we found this article and read it carefully. We will clearly and in detail explain the differences in working with them.
>
> 1) First, our approach is different. In fact, the f-mixup mentioned in [1] is similar to ours in that they both use the frequency domain to add disturbance information. But our process of generating triggers (adversarial instances in adversarial attacks) and the raw materials of triggers used are completely different. F-mixup generates candidates by replacing part of the mid- and high-frequency components of the clean examples with the corresponding frequency components of the reference image. That is to say, f-mixup first uses DFT to transform the image into the frequency domain, then uses a band-pass filter to extract part of the high-frequency components of the reference image, and then uses a band-stop filter to filter out the frequency components of the same part of the clean image. Then add part of the high-frequency components of the reference image to the spectrum of the clean image, and finally use IDFT to transform to the spatial domain to obtain the adversarial sample. Our frequency-based backdoor attack method selects part of the low-frequency components of the target category image as the raw material of the trigger, and then adds the trigger to the mid- and high-frequency components of the clean image through upsampling and other operations to generate a poisoned image. And in order to improve the generalization of CNN to poisoned images whose label is set as the target category, we randomly select the low-frequency components of the image from the target category as raw materials when generating each poisoned image.
>
> 2) Second, our purposes are different. The f-mixup in [1] is to destroy useful features in the image and introduce misleading information to perturb model prediction, thereby achieving adversarial attacks. Our attack method uses the low-frequency semantic information of the target category to inject the backdoor into the high-frequency component of the backdoor image in order to make the model better generalize to the trigger, thereby realizing the backdoor attack.
>
> [1] Li X C, Zhang X Y, Yin F, et al. Decision-based adversarial attack with frequency mixup[J]. IEEE Transactions on Information Forensics and Security, 2022, 17: 1038-1052.

---

> > ### Author Response · Authors · 2023-11-18
> > **Response to Reviewer 2gr1 (Part 2)**
> >
> > Q2: [2,3] has already provided comprehensive analyses about CNN's generalization in the frequency domain, I think most conclusions reached in Section 3 may also be derived from the previous work.
> >
> > A2: We carefully read and compared the two literatures you listed [2,3] and found that although some parts of our research looked similar, they were actually different. We will clearly list the viewpoints proposed by each and explain in detail the differences from their work, hoping to eliminate your doubts. A discussion of our work and their work has been added to Appendix A.8.
> >
> > 1) The work of [2] mainly analyzes the trade-offs between two data enhancement strategies, Gaussian enhancement and adversarial training, for different corruption types from the perspective of the frequency domain. They found that the augmentation strategy made the model tend to exploit low-frequency information in the input. This low-frequency bias reduces susceptibility to high-frequency damage and improves robustness to high-frequency damage while degrading performance against low-frequency damage. It is also proven that AutoAugment, a data augmentation, can play a key role in mitigating robustness issues. For adversarial perturbations, they proposed that while naturally trained models show higher concentrations of adversarial perturbations in the high-frequency domain, adversarial training biases these perturbations toward lower frequencies. And the perturbations in high frequencies revealed by natural training do not mean that removing high-frequency information from the input will produce a robust model.
> > 2) As mentioned in the paper, our work is inspired by [3] and attempts to explore the generalization of CNN to backdoor samples from the perspective of frequency domain. The contribution of [3] is that they first noticed the ability of CNN in capturing high-frequency components of images. [3] revealed from the data perspective that CNN achieves the trade-off between accuracy and robustness by utilizing the high-frequency components of images. They proposed that in generally labeled samples, the model first extracts low-frequency information and then gradually extracts high-frequency information to achieve higher training accuracy. In samples with chaotic labels, the model treats low-frequency and high-frequency information equally, which means that the model begins to choose to memorize the data. This part of the theory is also a hypothesis for our work. They also further explored the impact of training techniques on training results from a frequency perspective. For example, they found that a larger batch size has smaller changes in high-frequency information, narrowing the generalization gap, so it will lead to a smaller gap between training accuracy and test accuracy. BatchNorm can redistribute unbalanced frequency components. For adversarial learning, they found that adversarial learning has a smaller generalization gap, making the system convolution kernel smoother, and making a trade-off between accuracy and robustness.
> > 3) Our work not only analyzes the generalization mechanism of CNN to backdoor samples from the perspective of the frequency domain, but also studies the perturbations of some existing partial flip-flops in the frequency domain and the differences in the generalization performance of CNN to these frequency domain perturbations. . We found that effective triggers change the frequency domain distribution of samples, and CNN can generalize to a small number of poisoned samples by memorizing the frequency domain distribution changed by triggers. Our experiments also prove that the frequency domain distribution changed by the flip-flop is not always effective. For some flip-flops, only a part of it is effective. For some flip-flops, only a part of the perturbation can be used to achieve an efficient attack. Moreover, different frequency domain distributions of samples have different vulnerabilities to disturbances, and high-frequency components are more susceptible to disturbance than low-frequency components. In addition, after CNN establishes the association between feature distribution and labels, even if the triggering factors are diverse or unstable, backdoor attacks can be implemented as long as the frequency domain distribution is the same. In order to expand the usage scenarios of visible triggers, based on our proposed theory, we propose a method that can make visible triggers invisible, achieving invisibility while ensuring the performance of visible triggers. With the help of our theory, we also propose a backdoor attack algorithm based on low-frequency semantic information for target classes, achieving a high success rate in different data sets and models.

---

> > > ### Author Response · Authors · 2023-11-18
> > > **Response to Reviewer 2gr1 (Part 3)**
> > >
> > > Q2: [2,3] has already provided comprehensive analyses about CNN's generalization in the frequency domain, I think most conclusions reached in Section 3 may also be derived from the previous work.
> > >
> > > A2:Continuing from above. To sum up, our contribution is different from [2,3]. We use part of the conclusions of [3] as hypotheses, but this does not mean that our theory is that proposed by predecessors. We hope our explanation clears up your misunderstandings, if you have any further questions, please let us know and we will be happy to continue discussions on related work with you.
> > >
> > > Q3: The proposed attack is motivated by frequency domain analysis for CNNs , but ViT has different bias in the frequency domain [4], which means that the proposed method may not generalize to ViT. IMHO, this is a great limitation.
> > >
> > > A3：Our main work in this paper is to explore the memory mechanism of CNN for poisoned images from the perspective of frequency domain, and to study the generalization characteristics of CNN to perturbations generated by visible and invisible triggers in different frequency domains. The impact of different intensity perturbations of different frequency domain components on the generalization of CNN was also studied. The theoretical part does not cover the scope of Vit. Of course, we have noticed the difference in the attention of Vit and CNN to different components in the frequency domain proposed in the relevant work [5]. We are currently working hard to advance the work on Vit, but it is not the focus of the discussion of model generalization in this paper.
> > >
> > > Despite this, we still used our attack method to conduct relevant experiments on the Vit-base model, and the experimental results are shown in Tables 1 and 2 as below. It can be seen from the table that our stealth strategy and attack method are effective on the Vit model and can achieve high attack accuracy. Judging from the experimental results, the limitations you mentioned do not exist for our algorithm. We also put the relevant experimental results about Vit in Appendix A.6.
> > >
> > > Table 1: Backdoor attack on vision transformer
> > >
> > > |                                     | **BA**   | **ASR**    |
> > > |-------------------------------------|----------|------------|
> > > | BadNet                   | 94.78    | 93.77      |
> > > | Blended                | 96.54    | 99.68      |
> > > | LIRA                    | 17.52    | 87.87      |
> > > | TrojanNN                | 96.46    | 99.98      |
> > > | Ours($\rho=0.4$,$\varepsilon=\frac{1}{3}$) | **98.62** | **100.00** |
> > >
> > > Table 2: The Implementation of Invisible Strategies on vision transformer
> > >
> > > |         | **Origin-BA** | **Origin-ASR** | **Invisible-BA** | **Invisible-ASR** |
> > > |---------|---------------|----------------|------------------|-------------------|
> > > | BadNet  | 94.78         | 93.77          | 94.77            | 99.12             |
> > >
> > > [2] Yin D, Gontijo Lopes R, Shlens J, et al. A fourier perspective on model robustness in computer vision[J]. Advances in Neural Information Processing Systems, 2019, 32.
> > >
> > > [3] Wang H, Wu X, Huang Z, et al. High-frequency component helps explain the generalization of convolutional neural networks[C]//Proceedings of the IEEE/CVF conference on computer vision and pattern recognition. 2020: 8684-8694.
> > >
> > > [4]Park N, Kim S. How do vision transformers work?[J]. arXiv preprint arXiv:2202.06709, 2022.
> > >
> > > [5] Bai J, Yuan L, Xia S T, et al. Improving vision transformers by revisiting high-frequency components[C]//European Conference on Computer Vision. Cham: Springer Nature Switzerland, 2022: 1-18.

---

> > > > ### Author Response · Authors · 2023-11-18
> > > > **Response to Reviewer 2gr1 (Part 4)**
> > > >
> > > > Q4: The authors conducted experiments on MNIST, CIFAR-10 and Celeba, it is necessary to validate the proposed algorithms on ImageNet. In my experience, CNNs trained on ImageNet is far less sensitive to high frequencies than CNNs trained on the above datasets.
> > > >
> > > > A4：Thank you for pointing out our experimental shortcomings. To make up for the shortcomings, we conducted experiments on our related algorithms on ImageNet. However, due to GPU resource and time constraints, we currently use Tiny-ImageNet, which is widely used in the field of backdoor attacks. The dataset contains a subset of 200 classes, with 100,000 images for training (500 images per class) and 10,000 images for testing (50 images per class) [6]. We verified the backdoor attack (Table 3) and the stealth strategy (Table 4) with parameters set to (ρ = 0.4, ε = 1/3). We can see that our backdoor attack method and invisible strategy can still achieve good attack performance on ImageNet. The results obtained on ImageNet have been placed in Appendix A.5.
> > > >
> > > > Table 3: Backdoor attack on ImageNet
> > > >
> > > > |                                        | **BA** | **ASR** |
> > > > |----------------------------------------|--------|---------|
> > > > | LIRA                       | 0.05   | 99.85   |
> > > > | BPP                        | 58.14  | **100.00** |
> > > > | Blind                       | 1.97   | 45.21   |
> > > > | TrojanNN                   | 55.89  | 99.98   |
> > > > | Ours($\rho=0.4$,$\varepsilon=\frac{1}{3}$) | **59.11** | **100.00** |
> > > >
> > > > Table 4: The Implementation of Invisible Strategies on ImageNet
> > > >
> > > > |         | **Origin BA** | **Origin ASR** | **Invisible BA** | **Invisible ASR** |
> > > > |---------|---------------|----------------|------------------|-------------------|
> > > > | BadNet  | 65.31         | 97.17          | 61.18            | 99.16             |
> > > >
> > > >
> > > >
> > > >
> > > >
> > > > [6] Wu B, Chen H, Zhang M, et al. Backdoorbench: A comprehensive benchmark of backdoor learning[J]. Advances in Neural Information Processing Systems, 2022, 35: 10546-10559.

---

> ### Author Response · Authors · 2023-11-21
> **Further responses to your comments**
>
> Thank you for your reply. We have further clarified your question. We hope the new clarifications and results resolve your concerns. If you have any further questions, please ask and we'll be happy to continue answering your questions.
>
> Q1：the insufficient literature review still heavily weakens the motivation and novelty of this work.
>
> A1：Thank you sincerely for your suggestions. In fact, when you first asked the question, we had already conducted a relevant literature review in section 2.2 and listed some of the current frequency domain-based backdoor attack methods. The expression of these methods did not demonstrate our motivation very well. We are sorry that it did not meet your expectations, so we revised the deficiencies in section 1 and section 2.2 again to supplement the motivation and novelty of our theories and methods.
>
> 1) First of all, the current exploration of the mechanism of memorizing backdoor samples from the perspective of frequency domain is still lacking. Most frequency-based backdoor attack methods directly learn the CNN's memory ability of spectrum based on existing literature and make relevant assumptions, and then inject triggers based on this assumption. But they don’t know why setting the trigger in this way makes CNN generalize, and they don’t know the sensitivity of CNN to changes in the frequency components of each part. But we explored it. That is, we try to explain and explore the mechanism of CNN's memory of backdoor samples from the perspective of frequency domain. We further studied how the perturbations produced by visible and invisible triggers at different frequency components help the generalization ability of CNN.
>
> 2) Secondly, in order to make the backdoor attack in frequency more efficient and bypass the backdoor defense algorithm based on the spatial domain, and can effectively implement the backdoor attack on the target category while minimizing the destruction of the original image energy. We use the research conclusions obtained in section 3 to propose the backdoor attack algorithm shown in Figure 5(b). At the same time, it is considered that visible backdoor attacks have a high attack success rate, but because of the visibility of the trigger, their use is limited in scenarios that require high concealment. Therefore, combined with our theory, we propose a general invisible algorithm shown in Figure 5(a), which realizes the invisible of visible triggers, ensures the attack efficiency of visible triggers, and expands the usage scenarios of visible triggers.
>
> Q2：The algorithm shown in Figure 5(b) is similar to f-mixup in high-level ideas.
>
> A2:  As we replied before. The method we show in Figure 5(b) is completely different from the f-mixup method in terms of purpose and ideas. This also involves the purpose and difficulty of backdoor attack and adversarial attack tasks. Before comparing our two methods, it is necessary to correctly understand the difference between backdoor attacks and adversarial attacks. Moreover, just operating in the frequency domain does not mean that our high-level ideas are the same. Our purposes and methods are different, and the results we bring are also different. We will further explain the differences in methods to allay your concerns.
>
>
> 1) f-mixup destroys useful features in images and introduces misleading information to perturb model predictions, thereby achieving adversarial attacks. This purpose means that it does not need to consider outputting specific results, but only perturbs the output results of the model. At the same time, f-mixup is generated in a way that generates adversarial examples by replacing parts of the mid- and high-frequency components of the clean examples with the corresponding parts of the reference image. The images referenced in the generation of adversarial samples do not need to consider fixed categories, and the trigger raw materials are medium and high frequency components.
>
>
> 2) Our attack method utilizes the low-frequency semantic information of the target category and implants the backdoor in the high-frequency component of the backdoor image. This operation makes use of the CNN's memory mechanism for backdoor samples, so that the model can better generalize the triggers, thereby achieving backdoor attacks. And in order to improve the generalization of CNN to poisoned images whose label is the target category, we randomly select the low-frequency components of the image from the target category as raw materials when generating each poisoned image.

---

### Official Review · Reviewer_F6E6 · 2023-10-27

**Soundness:** 3 good
**Presentation:** 3 good
**Contribution:** 3 good
**Rating:** 6
**Confidence:** 2

**Summary:**

The study examines how triggers affect CNNs by looking at the frequency domain. The authors found that these triggers change the way images are distributed in this domain, which in turn affects CNN performance. They noted that higher frequencies are more vulnerable to attacks than lower ones. Based on these insights, they developed strategies to hide visible triggers and introduced a new backdoor attack method using low-frequency information. This method was effective against many defenses. The authors also provided ideas for future work, like using different triggers during training and testing. They hope their research encourages more exploration in the area of backdoor attacks and defenses.

**Strengths:**

1.  This paper adopt a novel perspective of learning the backdoor effect through the lens of frequency domain. Specifically, it is interesting to see how different frequency component affect the attack success rate of in current backdoor attack methods. This research provides an insightful understanding into the intricate dynamics between frequency components and the effectiveness of backdoor attacks.
2. The paper provides detailed experiments that comes with insightful conclusion, which is considered a good contribution to the backdoor community.

**Weaknesses:**

Two important works on frequency-based backdoor attack are missed:
[1] Zeng, Y., Park, W., Mao, Z.M. and Jia, R., 2021. Rethinking the backdoor attacks' triggers: A frequency perspective. In Proceedings of the IEEE/CVF international conference on computer vision (pp. 16473-16481).
[2] Feng, Y., Ma, B., Zhang, J., Zhao, S., Xia, Y. and Tao, D., FIBA: Frequency-Injection based Backdoor Attack in Medical Image Analysis Supplementary Material.
The paper should clearly state the differences between the current work and these previous works.

**Questions:**

Please refer to the weaknesses.

---

> ### Author Response · Authors · 2023-11-18
> **Response to Reviewer F6E6 (Part 1)**
>
> Thank you so much for your thoughtful comments and recognition of the significance of our work. We hope the following results and explanations address your concerns.
>
> We have carefully read the two important tasks you listed and we are very sorry for their omission. So below we will first introduce our contributions, and then clearly and in detail explain the differences from their work. A discussion of our work and their work has been added to Appendix A.8.
>
> Our work mainly explores the mechanism of CNN memory poisoning samples from the perspective of frequency domain. We analyze the changes in different frequency components of existing visible and invisible triggers with good performance. We found that CNN can generalize to poisoned samples by remembering the frequency domain distribution changed by triggers. We also explore the impact of triggering perturbations with different frequency domain components on the generalization of poisoning models for visible and invisible backdoor attacks, and demonstrate that high-frequency components are more susceptible to perturbations compared to low-frequency components. In addition, in order to improve the concealment of visible triggers in specific scenarios, we propose a general strategy to hide visible triggers. We also propose a backdoor attack algorithm based on low-frequency semantic information, which is proven to be effective and can bypass multiple backdoor defense methods.
>
> Literature [1] also adds triggers in the frequency domain. What we have in common is that we all use the low-frequency information of the target category image to introduce the semantics of the target category into the poisoned image. But our methods of generating triggers and specific training are completely different.
>
> Literature [1] uses FFT to transform the image from the spatial domain to the frequency domain, and then linearly adds the low-frequency component of the clean image and the low-frequency component of the target image through a certain ratio as the low-frequency component of the poisoned image, thereby achieving frequency injection of the trigger. Literature [1] uses FFT to transform the image from the spatial domain to the frequency domain, and then linearly adds the low-frequency component of the clean image and the target image through a certain ratio as the low-frequency component of the poisoned image, thereby achieving frequency injection of the trigger. And their training process introduced noise images similar to those in the literature [2] to enhance the uniqueness of the trigger. However, our frequency-based backdoor attack method selects part of the low-frequency components of the target category image as the raw material of the trigger. The poisoned image is then generated by adding triggers to the mid- and high-frequency components of the clean image through operations such as upsampling. And in order to improve the generalization of CNN to poisoned images whose label is the target category, we randomly select the low-frequency components of the image from the target category as raw materials when generating each poisoned image, and do not use noise images.
>
>
> [1] Feng, Y., Ma, B., Zhang, J., Zhao, S., Xia, Y. and Tao, D., FIBA: Frequency-Injection based Backdoor Attack in Medical Image Analysis Supplementary Material. The paper should clearly state the differences between the current work and these previous works.
>
> [2] Tuan Anh Nguyen and Anh Tuan Tran. Wanet - imperceptible warping-based backdoor attack. In ICLR, 2021.

---

> > ### Author Response · Authors · 2023-11-18
> > **Response to Reviewer F6E6 (Part 2)**
> >
> > In fact, although we all explore the characteristics of triggers from the perspective of frequency domain, our work is completely different from that of literature [3].
> >
> > 1)  [3] studied the spectrum of poisoned images and showed that most triggers produce artifacts in the high-frequency components of input images of large sizes, and explained the reasons for this phenomenon accordingly. That is, the high-frequency artifacts of local patching mainly originate from the high-frequency components of the trigger pattern itself, while the high-frequency artifacts of large-size or global patching originate from the trigger pattern itself or the method of inserting the trigger. GAN-generated backdoor data originates from the upsampling operation used by GAN. But the difference is that our work is inspired by [4] and explains the mechanism of CNN memory triggers from the frequency domain perspective (section 3). We propose and prove that CNN can memorize specific frequency domain feature distributions and establish the association between feature distributions and labels. Therefore, backdoor attacks can be achieved even if the trigger factors are diverse or not fixed. We used this mechanism to explore the impact of perturbations produced by visible and invisible triggers in different frequency domain components on the generalization ability of CNN, as well as the differences in the generalization ability of CNN to different perturbation intensities imposed on different frequency domain components. And it is proved that high-frequency components are more sensitive to disturbances than low-frequency components.
> >
> > 2) Reference [3], based on their identified discovery that triggers in large-sized input images generate high-frequency artifacts, developed a detector specifically designed for frequency artifacts. They trained this detector using poisoned data to ensure its generalization to other poisoned samples. [3] also created a smooth trigger for backdoor attacks. Their generation of smooth triggers is generalized as a two-layer optimization problem, and neither the generated triggers themselves nor the final patch image contain any high-frequency components. Our work is based on the exploration of the impact of perturbations produced by existing triggers in different frequency domain components on the generalization ability of CNN, and proposes a general strategy for making visible triggers invisible. It makes the trigger invisible by removing the low-frequency components of the trigger. We also proposed a backdoor attack algorithm based on low-frequency semantic information for target classes based on the mechanism of CNN memory triggers, achieving a high success rate in different data sets and models.
> >
> > 3) In fact, the high-frequency artifacts proposed in the literature [3] are only obvious in large-size input images (224\*224), and are difficult to observe for smaller-size (32\*32) images. It therefore has limitations for the defense of our approach. And the smooth trigger they proposed only adds triggers at low frequencies, which will cause the poisoned image to be abnormally obvious in the spatial domain.
> >
> > [3] Zeng, Y., Park, W., Mao, Z.M. and Jia, R., 2021. Rethinking the backdoor attacks' triggers: A frequency perspective. In Proceedings of the IEEE/CVF international conference on computer vision (pp. 16473-16481).
> >
> > [4] Haohan Wang, Xindi Wu, Zeyi Huang, and Eric P Xing. High-frequency component helps explain the generalization of convolutional neural networks. In Proceedings of the IEEE/CVF conference on computer vision and pattern recognition, pp. 8684–8694, 2020.

---

### Official Review · Reviewer_Uoqj · 2023-10-29

**Soundness:** 2 fair
**Presentation:** 3 good
**Contribution:** 3 good
**Rating:** 6
**Confidence:** 3

**Summary:**

This paper propose to study the backdoor attack on CNN in frequency domain. It shows that high frequency component are more susceptible to perturbations. It further proposes a strategy for rendering visible backdoor attack invisible and proposes a backdoor attack algorithm based on low-frequency component from target class.

**Strengths:**

1). It is interesting and novel to study the backdoor attack on CNN in frequency domain. The proposed algorithm utilizing low-frequency component from target class is also interesting.

2). Overall, the paper is clear and well-written.

**Weaknesses:**

1). The experiments is not enough to show the effectiveness of the proposed method against defense. For backdoor attacks, it is easy to achieve high ASR with no defense. Though empirical evidences have been provided in Fig.6, I still have doubts on the effectiveness of the propsoed method against defense. It would be more convincing to report the result against some defense methods e.g. the defense methods supported by backdoorbench [1].

2). For the strategy rendering visible backdoor attack invisible, it lacks comparison with other invisible backdoor attacks. Since invisible backdoor attacks mainly rely on the perturbation on high frequency components, what is the relation or difference between visible backdoor attack and invisible backdoor attack after masking the low-frequency perturbation?

3). Fig.4 shows that it requires smaller perturbation on high frequency than on low frequency to achieve high attack success rate. However, when comparing to the original image, since most mass concentrate in low-frequency components, the small perturbation on high frequency might be relatively large comparing to the original image. It might require more results to show that high frequency components are more susceptible to backdoor attack

Overall, I think the experiments in this paper is not sufficient to support the analysis in this paper and verify the effectiveness of the proposed method.

[1] Baoyuan Wu, Hongrui Chen, Mingda Zhang, Zihao Zhu, Shaokui Wei, Danni Yuan, Chao Shen. BackdoorBench: A Comprehensive Benchmark of Backdoor Learning. NeurIPS 2022.

**Questions:**

As mentioned in the weakness section, I have several questions regarding the experiments in this paper.

1). When rendering visible backdoor attack invisible, what is the relation between the visible backdoor attacks and invisible backdoor attacks?

2). For both the rendering strategy and the proposed backdoor attack algorithm, are they still be effective facing conventional defence methods?

3). It has already been observed that high frequency components are more susceptible to attacks such as adversarial attacks. Is there any new takeaways regarding the backdoor attack?

---

> ### Author Response · Authors · 2023-11-18
> **Response to Reviewer Uoqj (Part 1)**
>
> Thank you very much for your time and insightful comments. We are encouraged that our work is recognized. We hope the following new clarifications and results can address your concerns. We are happy to answer more questions and conduct more experiments if needed.
>
> Q1：When rendering visible backdoor attack invisible, what is the relation between the visible backdoor attacks and invisible backdoor attacks?
>
> A1: In this work, we achieve the invisibility of visible backdoor attacks from the frequency domain. In fact, we modify the visible trigger in the frequency domain to make it invisible and still maintain good offensiveness. It can be said that the generated invisible trigger uses a visible trigger with good performance as raw material, and uses the high-frequency disturbance of the visible trigger to implement a backdoor attack. The attack performance of the visible trigger will also affect the attack performance of the generated invisible trigger to a certain extent.
> 	The invisible strategy we propose makes up for the shortcomings of most existing visible backdoor attacks, that is, most visible backdoor attacks [1, 2, 3] have good attack performance but poor stealth, making them Application scenarios are limited. Using the invisible strategy we proposed can help visible triggers be applied in scenarios with high stealth requirements.
>
> Q2：For  the proposed backdoor attack algorithm, are they still be effective facing conventional defence methods?
>
> A2：Thanks for pointing this out. We used GradCam, Fine-Pruning and STRIP to evaluate backdoor defense in our original paper submission, and the results proved that our frequency domain-based backdoor attack algorithm is effective in combating backdoor defense (Figure 6). Now we have added three new backdoor defense methods, NC [4], DBD [5] and CLP [6], to further evaluate the frequency domain-based attack algorithm we proposed respectively. The experimental results are shown in Tables 1 and 2 as below. Experimental results show that although the performance of our attack algorithm is slightly degraded on some defense algorithms, it is still effective. We have added this part to Appendix A.7.
>
> Table 1: Mitigation results of backdoor defense methods DBD and NC on our algorithm
>
> |                                    | **DBD-BA** | **DBD-ASR** | **NC-BA** | **NC-ASR** |
> |------------------------------------|------------|-------------|-----------|------------|
> | BadNet [1]                  | 89.65      | 1.28        | 89.05     | 1.27       |
> | Blended[7]                | 69.91      | 99.68       | 93.47     | 99.92      |
> | WaNet[8]                    | 80.9       | 6.61        | 91.80     | 7.53       |
> | ISSBA[9]                  | 63.5       | 99.51       | 90.99     | 0.58       |
> | Ours($\rho=0.4$,$\varepsilon=\frac{1}{3}$) | 70.29  | 100.00      | 94.26     | 91.80      |
>
> Table 2 Mitigation results of our algorithm by backdoor defense method CLP
>
> |                                    | **CLP-BA** | **CLP-ASR** |
> |------------------------------------|------------|-------------|
> | IAD[2]                     | 90.3       | 2.17        |
> | Blended[7]                 | 91.32      | 99.74       |
> | WaNet[8]                    | 81.91      | 78.42       |
> | ISSBA[9]                   | 91.38      | 68.13       |
> | Ours($\rho=0.4$,$\varepsilon=\frac{1}{3}$) | 94.55 | 98.41       |
>
> [1] Gu T, Dolan-Gavitt B, Garg S. Badnets: Identifying vulnerabilities in the machine learning model supply chain[J]. arXiv preprint arXiv:1708.06733, 2017.
>
> [2] Nguyen T A, Tran A. Input-aware dynamic backdoor attack[J]. Advances in Neural Information Processing Systems, 2020, 33: 3454-3464.
>
> [3] Bagdasaryan E, Shmatikov V. Blind backdoors in deep learning models[C]//30th USENIX Security Symposium (USENIX Security 21). 2021: 1505-1521.
>
> [4] Wang B, Yao Y, Shan S, et al. Neural cleanse: Identifying and mitigating backdoor attacks in neural networks[C]//2019 IEEE Symposium on Security and Privacy (SP). IEEE, 2019: 707-723.
>
> [5] Huang K, Li Y, Wu B, et al. Backdoor defense via decoupling the training process[J]. arXiv preprint arXiv:2202.03423, 2022.
>
> [6] Zheng R, Tang R, Li J, et al. Data-free backdoor removal based on channel lipschitzness[C]//European Conference on Computer Vision. Cham: Springer Nature Switzerland, 2022: 175-191.
>
> [7] Chen X, Liu C, Li B, et al. Targeted backdoor attacks on deep learning systems using data poisoning[J]. arXiv preprint arXiv:1712.05526, 2017.
>
> [8]Nguyen A, Tran A. Wanet--imperceptible warping-based backdoor attack[J]. arXiv preprint arXiv:2102.10369, 2021.
>
> [9]Li Y, Li Y, Wu B, et al. Invisible backdoor attack with sample-specific triggers[C]//Proceedings of the IEEE/CVF international conference on computer vision. 2021: 16463-16472.

---

> > ### Author Response · Authors · 2023-11-18
> > **Response to Reviewer Uoqj (Part 2)**
> >
> > Q3：It has already been observed that high frequency components are more susceptible to attacks such as adversarial attacks. Is there any new takeaways regarding the backdoor attack?
> >
> > A3：As we can see, we discussed the impact of current triggers on both low-frequency and high-frequency components, as well as the difference in generalization of CNNs to backdoor samples. We also demonstrated the vulnerability of high-frequency components. To address this issue, we are actively engaged in conducting relevant extended research efforts. For example,
> >
> > 1) The first is the defense problem based on the frequency domain. There are already some frequency-specific backdoor defense methods. Most of the current frequency-based defenses are the detection of backdoor samples [10], or it is difficult for the algorithm to restore the backdoor samples to a clean state and maintain the accuracy of the clean image [11], but our idea is different. In this paper, we have confirmed that when some triggers do not reach a certain perturbation intensity at high or low frequencies, CNN will not remember these triggers, so we are trying to detect and weaken the backdoor perturbations in the frequency domain to achieve filtering of backdoor samples. and recovery.
> >
> > 2) At present, physical attacks are easier to apply in actual scenarios [12], but it is difficult for physical attack triggers to be invisible during the training process. Therefore, we consider implanting a backdoor by adding small perturbations in the frequency domain during the training phase, and visible triggers that can achieve corresponding attack effects are generated and applied in the inference phase to enhance the invisibility of physical backdoor attacks in the training phase. This method is theoretically feasible because according to the linear convolution properties of DCT, the triggers added in the spectrum should be separable in the spatial domain.
> >
> >
> >
> >
> > [10] Al Kader Hammoud H A, Bibi A, Torr P H S, et al. Don't FREAK Out: A Frequency-Inspired Approach to Detecting Backdoor Poisoned Samples in DNNs[C]//Proceedings of the IEEE/CVF Conference on Computer Vision and Pattern Recognition. 2023: 2337-2344.
> >
> > [11] Hammoud H A A K, Ghanem B. Check Your Other Door! Creating Backdoor Attacks in the Frequency Domain[J]. arXiv preprint arXiv:2109.05507, 2021.
> >
> > [12] Wenger E, Passananti J, Bhagoji A N, et al. Backdoor attacks against deep learning systems in the physical world[C]//Proceedings of the IEEE/CVF Conference on Computer Vision and Pattern Recognition. 2021: 6206-6215.

---

> > > ### Author Response · Authors · 2023-11-18
> > > **Response to Reviewer Uoqj (Part 3)**
> > >
> > > Regarding other weaknesses raised by reviewers.
> > >
> > > Q4：It might require more results to show that high frequency components are more susceptible to backdoor attack
> > >
> > > A4: I understand your concern regarding the accuracy of measuring the difficulty of attacks due to the varying information content in different frequency components. Indeed, low-frequency components contain a relatively greater amount of information, which is an important aspect we have considered in our paper.
> > > 	In our paper, we analyzed the difference in generalization of CNN to perturbations produced by visible and invisible triggers in the frequency domain, and then further explored the difference in generalization of CNN to different perturbation strengths of different frequency domain components. Our comparison in the paper emphasizes the sensitivity of high-frequency components to perturbations and the conditions for frequency domain perturbations required to achieve efficient attacks. And the purpose of our experiment was to better account for this when adding triggers or weakening existing ones. Regarding your proposal to study the generalization characteristics of CNN based on changes in the relative intensity of frequency domain perturbations, we did think about it in our initial research, but we did not explore it out of consideration for inspiration for backdoor attacks and defenses. . We believe that the existing experimental results already support our conclusions, and we must take into account the resource-intensive nature of additional experiments and the time constraints at this stage. Nonetheless, we remain committed to advancing our research. If you still feel that it is necessary to study the relative perturbation intensity of different frequency components, we are very happy to design experiments to further study the generalization properties of CNN as a result of changes in the relative intensity of perturbations in the frequency domain. And we promise to supplement our findings with more experimental results in future submissions.

---

> ### Comment · Reviewer_Uoqj · 2023-11-21
> **Thanks for the response**
>
> I thank author for the detailed response. The experimental results against defense methods have addressed my concerns over the effectiveness of the proposed methods. Therefore I am rising my rating to 6.

---

> > ### Author Response · Authors · 2023-11-21
> > **Sincere thanks to reviewer  Uoqj**
> >
> > We are very happy to solve your problem well and thank you for your encouragement and improving your score!

---

### Author Response · Authors · 2023-11-21
**Looking forward to discussions before deadline**

Dear reviewers:

As the deadline for revisions and discussions approaches, we would like to follow up on your feedback. We understand that your schedule may be busy, but we believe we have addressed all of your concerns. If you still have questions or are unclear after reading our response, please let us know and we'll be happy to provide clarification and discussion.

Best regards and thanks,

Authors of #912

---

### Author Response · Authors · 2023-11-22

We are grateful to the reviewers for their insightful and constructive feedback. We are pleased to see the reviewers' recognition of our work and the writing of the paper. We are very happy to see that the study of CNN's generalization of backdoors in the frequency domain is novel and interesting, a good contribution (Reviewer Uoqj, F6E6, 6jrw), has good motivation (Reviewer 2gr1), the paper results can support the insightful conclusions (Reviewer F6E6, 6jrw), the paper is clear and well written (Reviewer Uoqj, F6E6, 2gr1, 6jrw).

At the same time, we are very grateful to all the reviewers for taking their precious time to review our paper, and some reviewers for following up on our responses. We carefully responded to the questions raised by the reviewers in detail one by one, made detailed comparisons with other people's work, and also added relevant experiments. We also further responded seriously to some reviewers’ questions during the rebuttle period. Constructive suggestions help improve our manuscript and expand its contribution to the field. We carefully considered and addressed every point raised, which ensured our research was on a more solid footing. For the revision of the paper, the main parts are as follows:

1) **ABSTRACT& CONCLUSION**: Express our work and contributions more clearly

2) **INTRODUCTION**: Express our work and contributions more clearly, re-explain the motivation of the algorithm, and enhance understanding

3) **RELATED WORK**: An enriched review of related backdoor attacks in the frequency domain and a restatement of the motivations for our research

4) **APPENDIX**: A.5 adds the verification of invisible strategies and attack algorithms on ImageNet, and the relevant results are added to tables 5 and 6. A.6 adds the verification of invisible strategies and attack algorithms on the vision transformer, and the relevant results are added to tables 7 and 8. A.7 adds attack experiments based on other backdoor defense methods, and the relevant results are added to tables 9 and 10. The above experimental results all demonstrate the robustness and effectiveness of our algorithm. A.8 adds a discussion of differences from other backdoor attack efforts.

---

### Meta-Review · Area_Chair_k2XA · 2023-12-06

**Metareview:**

The paper analyzes CNN's generalization of backdoors in the frequency domain and propose a novel backdoor attack method based on their finding.

Strengths:

1. The authors explored the mechanism of CNN's memory of backdoor images and try to explore their generalization from the perspective of the frequency domain and further studied the difference between visible and invisible triggers from the frequency perspective.  Their work can give the follower researchers a new perspective on backdoor's generalization theory.

2. They also propose a new low-frequency backdoor, which can work efficiently against many backdoor defense methods.

Weaknesses:

1. The findings can only be implemented in CNN, and cannot get a generalized backdoor performance on ViT and other models. Thus, their conclusions are not universal for all model architectures.

Decision:

Most reviewers show positive attitudes toward this paper. Therefore, I decided to accept this paper as a poster.

**Justification For Why Not Higher Score:**

Can only be implemented in CNN, and cannot get a generalized backdoor performance on ViT and other models.

**Justification For Why Not Lower Score:**

The perspectives are novel and interesting and their results can support the insightful conclusions.

---

### Decision · Program_Chairs · 2024-01-16

Accept (poster)